# Beyond Confidence: Reliable Models Should Also Consider Atypicality

**Mert Yuksekgonul**
Stanford University
merty@stanford.edu

**Linjun Zhang**
Rutgers University
lz412@stat.rutgers.edu

**James Zou**[‡]
Stanford University
jamesz@stanford.edu

**Carlos Ernesto Guestrin**[‡]
Stanford University, CZ Biohub
guestrin@stanford.edu

## Abstract

While most machine learning models can provide confidence in their predictions, confidence is insufficient to understand a prediction's reliability. For instance, the model may have a low confidence prediction if the input is not well-represented in the training dataset or if the input is inherently ambiguous. In this work, we investigate the relationship between how atypical (rare) a sample or a class is and the reliability of a model's predictions. We first demonstrate that atypicality is strongly related to miscalibration and accuracy. In particular, we empirically show that predictions for atypical inputs or atypical classes are more overconfident and have lower accuracy. Using these insights, we show incorporating atypicality improves uncertainty quantification and model performance for discriminative neural networks and large language models. In a case study, we show that using atypicality improves the performance of a skin lesion classifier across different skin tone groups without having access to the group attributes. Overall, *we propose that models should use not only confidence but also atypicality to improve uncertainty quantification and performance*. Our results demonstrate that simple post-hoc atypicality estimators can provide significant value.[1]

## 1 Introduction

*Typicality* is an item's resemblance to other category members [RM75]. For example, while a dove and a sparrow are typical birds, a penguin is an atypical bird. Many works from cognitive science (e.g., [Rip89, RSS73, MP80]) suggest that typicality plays a crucial role in category understanding. For instance, humans have been shown to learn, remember, and refer to typical items faster [Mur04]. Similarly, the representativeness heuristic is the tendency of humans to use the typicality of an event as a basis for decisions [TK74]. This cognitive bias is effective for making swift decisions, but it can lead to poor judgments of uncertainty. For instance, the likelihood of typical events can be overestimated [TK74] or uncertainty judgments can be inferior for atypical events [TK92].

While it is hard to quantify the uncertainty of human judgments, machine learning models provide confidence in their predictions. However, confidence alone can be insufficient to understand the reliability of a prediction. For instance, a low-confidence prediction could arise from an ambiguity that is easily communicated, or due to the sample being underrepresented in the training distribution.

---

[1]Our code is available at https://github.com/mertyg/beyond-confidence-atypicality
[‡]Joint Advisors.

37th Conference on Neural Information Processing Systems (NeurIPS 2023).

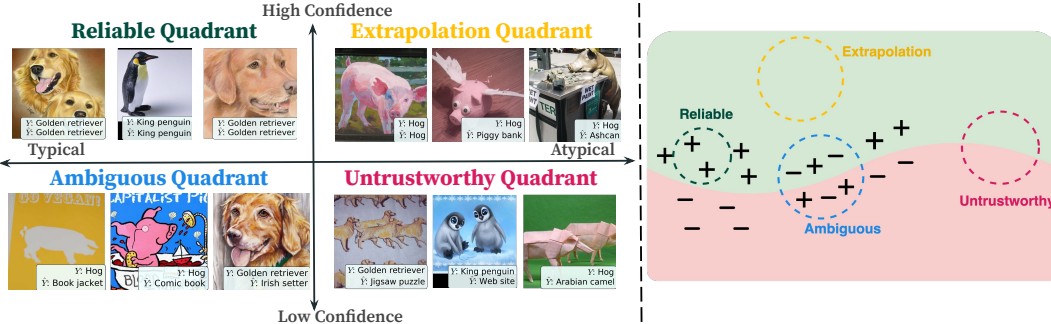

Figure 1: **Atypicality in Uncertainty. Left:** We show examples from the ImageNet-R dataset with our atypicality framework. **Right:** We provide a conceptualization of the quadrants. Using atypicality, we can understand prediction quality (§3), improve predictions (§4), and prediction sets (§5).

Similarly, a high-confidence prediction could be reliable or miscalibrated. Our main proposal is that *models should quantify not only the confidence but also the atypicality* to understand the reliability of predictions or the coverage of the training distribution. However, many machine learning applications rely on pretrained models that solely provide confidence levels, devoid of any measure of atypicality.

**Contributions:** To support our position, we use a simple formalization of atypicality estimation. With the following studies, we show that by using simple atypicality estimators, we can:

**1. Understand Prediction Quality:** Calibration is a measure that assesses the alignment between predicted probabilities of a model and the true likelihoods of outcomes [GR07]. Neural networks [GPSW17] or even logistic regression [BMWX21a] can be miscalibrated out-of-the-box. Here, we argue that using atypicality can give insights into when a model's confidence is reliable. Through theoretical analysis and extensive experimentation, we demonstrate that atypicality results in lower-quality predictions. Specifically, *we show that predictions for atypical inputs and samples from atypical classes are **more overconfident and have lower accuracy***.

**2. Improve Calibration and Accuracy:** *Recalibration* methods offer some mitigation to miscalibration [GPSW17] by adjusting a probabilistic model. We show that models need different adjustments according to the atypicality of inputs and classes, and atypicality is a key factor in recalibration. In light of these findings, we propose a simple method: *Atypicality-Aware Recalibration.* Our recalibration algorithm takes into account the atypicality of the inputs and classes and is simple to implement. We show that complementing recalibration methods with atypicality improves uncertainty quantification and the accuracy of predictors. Further, in a case study for skin lesion classification, we show that atypicality awareness can improve performance across different skin-tone subgroups without access to group annotations.

**3. Improve Prediction sets:** An alternative approach to quantify uncertainty is to provide prediction sets that contain the label with high probability [ABMJ20]. Here, we investigate existing methods with atypicality and show that prediction sets could underperform for atypical or low-confidence samples. By using atypicality, we demonstrate the potential for improving prediction sets.

Overall, we propose that **models should also consider atypicality, and we show simple- and easy-to-implement atypicality estimators can provide significant value**.

## 2 Interpreting Uncertainty with Atypicality

**Motivation:** In many machine learning applications, we have access to a model's confidence, which aims to quantify the likelihood that a prediction will be accurate. In classification, model output is a probability distribution over classes and confidence is the predicted probability of the top class, i.e. $\max_y \hat{\mathbb{P}}(Y = y | X = x)$. In practical scenarios, confidence is the primary tool used to evaluate the reliability of a prediction where higher confidence is associated with better predictions. However, the uncertainty in confidence can stem from different sources that require different treatment [MKvA$^+$21].

Here, we call a prediction *reliable* if it is high-confidence and well-calibrated. High confidence could be reliable or miscalibrated, and low confidence could be due to ambiguity or rare inputs. We propose that *atypicality* provides a natural way to understand reliability when combined with confidence. A sample is called typical if it is well-represented in the previously observed samples, e.g., an image of a dog that is similar to other dogs in the training data. However, if the image is unlike any other seen during training, it is atypical. We argue that atypicality can help us interpret a prediction's reliability. Below we categorize samples and predictions according to atypicality and confidence in four quadrants (Figure 1).

**High-confidence and representative:** Reliable predictions often fall within the **Reliable Quadrant**, which includes *typical, high-confidence* samples. These samples are well-represented in the training dataset (typical), thus we expect the high-confidence prediction to be reliable. For instance, the first image on the top left (Figure 1) is a typical golden retriever and the model makes a reliable prediction.

**High-confidence yet far from the support:** Having high-confidence does not always indicate reliability. If the sample does not have support in the training distribution, the confidence could be miscalibrated. Such samples lie in the **Extrapolation Quadrant** which contains *atypical, high-confidence* samples. For instance, the second image in the top right of Figure 1 is a *toy* hog and the model has not seen similar ones during training.

**Low confidence due to ambiguity:** In contrast, low confidence could also be reliable when it correctly reflects an ambiguity. Such samples are in the **Ambiguous Quadrant** that contains *typical, low-confidence* samples. These are typical since they may represent multiple classes; yet, due to ambiguity, the model's confidence is low. For instance, the second image in the bottom left of Figure 1 can both be a hog and a comic book.

**Low confidence and rare:** For samples that are not well-represented in training data, we expect to have low-quality predictions. **Untrustworthy Quadrant** comprises *atypical, low-confidence* samples that can include extremely rare subgroups, for which we expect miscalibration and lower accuracy. For example, the image in Figure 1 bottom right is an origami hog that was not seen in training.

These examples suggest that relying solely on confidence does not provide a complete understanding of the reliability of the predictions, and we can use atypicality to interpret and improve reliability.

**Formalizing Atypicality:** Atypicality here is defined with respect to the training distribution. Informally, an input or a class is atypical if it is not *well-represented* in the training distribution. For instance, if there are no or limited similar examples to an input, it can be called atypical. Note that this notion is not restricted to being 'out-of-distribution' [HG16a], since in-distribution groups could also be atypical or rare, and our goal is to perform reliably for the entire spectrum.

Formally, let $X \in \mathbb{R}^d$ be the random variable denoting features and $Y \in \mathcal{Y} = \{1, 2, ..., C\}$ denote the class, where we focus on classification.

**Definition 2.1** (Input Atypicality). We define the atypicality of the input $x$ as[2]

$$a_X(x) = -\max_y \log \mathbb{P}(X = x | Y = y).$$

We use the logarithm of the class-conditional densities due to high dimensionality and density values being close to zero. Intuitively, for a dog image $x$, if $\mathbb{P}(X = x | Y = \text{dog})$ has a low value, we call $x$ an atypical dog image. Overall, if $a(x)$ is high, then we call $x$ an atypical input. Specifically, if an input is not typical for any class, then it is atypical with respect to the training distribution. Similarly, we can also use marginal density, $\mathbb{P}(X = x)$, or distance[3] to quantify atypicality.

Similarly, the notion of atypical (rare) classes is prevalent in imbalanced classification [CWG+19, ZCLJ21]. Ensuring reliable performance for atypical classes can be safety-critical, e.g., for a rare presence of dangerous melanoma [DVN+22]. We define class atypicality in the following:

**Definition 2.2** (Class Atypicality). For a class $y$, atypicality of a class is defined as

$$a_Y(y) = -\log \mathbb{P}(Y = y).[4]$$

---

[2]Here atypicality differs from 'typical sets' in information theory that refers to a sequence of variables [TJ06].

[3]For an input $x$, if the nearest neighbor (NN) distance is large, then we call $x$ atypical as all inputs in the training set are far from $x$. Density and distance are connected through non-parametric density estimation and [JKGG18] shows that NN distance can recover high-density regions.

[4]When the meaning is unambiguous, we omit the subscript to denote $a(X)$ or $a(Y)$ for notational brevity.

**Estimating Atypicality for Discriminative Models:** Quantifying input atypicality requires access to the class-conditional / marginal distributions. In practice, for neural networks trained for classification, these distributions are unavailable and we need to perform the estimation. This estimation can be challenging if the dimensionality is large, or the data is unstructured, requiring assumptions about the distributions. Prior works [MKvA+21, LLLS18] showed that Gaussian Mixture Models (GMMs) in the embedding space of neural networks can be used to model these distributions.

In experiments, we use Gaussians with shared covariance, i.e. $\hat{\mathbb{P}}(X = x | Y = c) \sim N(\hat{\mu}_c, \hat{\Sigma})$, to estimate input atypicality. We perform the estimation in the penultimate layer of neural networks used to make predictions, using maximum-likelihood estimation with samples from the training data. We explore other metrics, such as $k$-Nearest Neighbors distance. We give implementation details and results with different metrics in Appendix A.1. With these estimators, atypicality estimation is cheap and can run on a CPU. Our goal is to show that simple estimators can already reap large benefits. Our framework is flexible and exploring more sophisticated estimators is a topic for future work.

**Atypicality for LLMs:** LLMs are increasingly used for classification [BMR+20]. Modern LLMs are autoregressive models that compute a marginal distribution, $\hat{\mathbb{P}}_{\text{LLM}}(X)$. We compute the negative log-likelihood of a prompt or a label and use this as an atypicality metric, i.e. $a_X(x) = -\log \hat{\mathbb{P}}_{\text{LLM}}(x)$, $a_Y(y) = -\log \hat{\mathbb{P}}_{\text{LLM}}(y)$. Similar to the discriminative setting, atypicality here aims to quantify whether a prompt is well-represented in the training data. A larger value for $a(X)$ would imply that a prompt is more atypical. Below, we present typical and atypical prompts for the AGNews dataset:

---

**Classify the news articles into the categories of World, Sports, Business, and Technology.**
**Article:** Safin tallest obstacle to host #39;s patriotic games hope AS tennis fans go, Houston #39;s Jim #39;Mattress Mack #39; McIngvale is very rich, extremely forthright, exceedingly patriotic and unflinchingly Republican.
**Answer:** *Atypicality:* 353.45, *Percentile:* %94.5

**Classify the news articles into the categories of World, Sports, Business, and Technology.**
**Article:** Delta Air Lines Prepares Chapter 11 Filing Delta Air Lines Inc. could file for Chapter 11 bankruptcy protection as soon as next week, a source familiar with the matter said yesterday.
**Answer:** *Atypicality:* 171.50. *Percentile:* %0.9

---

# 3 Understanding the Prediction Quality with Atypicality

In this section, we show how our framework can be applied to understand the quality of predictions. **Experimental Setup:** We investigate three classification settings across a range of datasets:

1. **Balanced Supervised Classification:** We use ResNet18-50-152 [HZRS16], WideRes-Net28 [ZK16], RoBERTa [LOG+19] trained on ImageNet [DDS+09], CIFAR10,100 [Kri09], MNLI [WNB18] respectively.
2. **Imbalanced Supervised Classification:** We use ResNet18, ResNext50, ResNet152 trained on CIFAR-LT, ImageNet-LT and Places365-LT where models and data are mostly from [ZCLJ21, MKS+20]. Note that all of the test and validation sets have balanced class distributions.
3. **Classification with LLMs:** We use open-source Alpaca7B [TGZ+23] on IMDB [MDP+11], TREC [LR02], and AG News [ZZL15] datasets with the prompts from [ZWF+21].

Details on datasets, models, and prompts are in Appendix B. Our experiments were run on a single NVIDIA A100-80GB GPU. We report error bars over 10 random calibration/test splits.

## 3.1 Atypicality is Correlated with Miscalibration

We first explore the importance of atypicality to understand model calibration. Calibration quantifies the quality of a probabilistic model [GR07]. Informally, a model is considered perfectly calibrated if all events that are predicted to occur $P\%$ of the time occur $P\%$ of the time for any $P \in [0, 100]$.

For the sake of simplicity, consider a binary classification problem where the predictor is $\hat{\mathbb{P}} : \mathcal{X} \rightarrow [0, 1]$. We quantify miscalibration with Calibration Error (CE):

$$\text{CE}[\hat{\mathbb{P}}] = \mathbb{E}[|\mathbb{P}(Y | \hat{\mathbb{P}}(X) = p) - p|].$$

It is computationally infeasible to calculate the above expectation with the conditional probability $\mathbb{P}(Y | \hat{\mathbb{P}}(X) = p)$. In practice, we use a binned version of this quantity, Expected Calibration Error (ECE) [NCH15, GPSW17], to estimate CE. See Appendix C.1 for a formal definition.

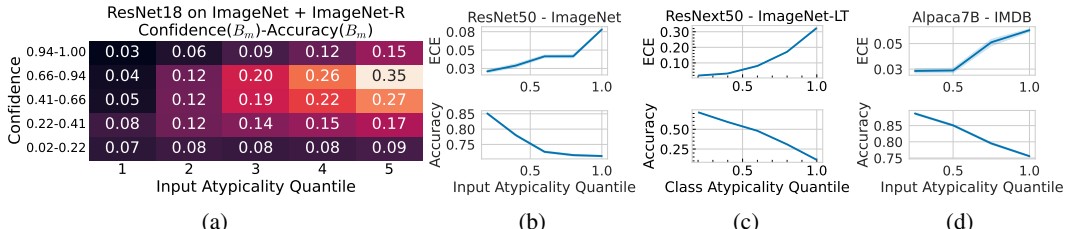

(a)          (b)          (c)          (d)

Figure 2: **Atypical Samples Have Low-Quality Predictions. (a)** Here, samples are grouped according to the Input Atypicality (x-axis) and Confidence (y-axis), to the right meaning more atypical. Values show the difference between confidence and accuracy, lighter color indicates more overconfidence. Within the same confidence range, atypical groups have more miscalibration and are more overconfident. **(b,c,d)** Predictions for atypical samples are less accurate and more miscalibrated in balanced and imbalanced supervised classification and classification with LLMs.

Here, we aim to examine the relationship between model calibration and atypicality. Given any $K > 1$, we consider the quantiles of $a(X)$, $a_1, a_2, \ldots, a_{K+1}$ such that $\mathbb{P}(a(X) \in (a_k, a_{k+1})) = 1/K$ for $k \in [K]$. For imbalanced classification problems, we compute the quantiles using the class atypicality. Specifically, we investigate the atypicality-conditional calibration error $\text{ECE}[\hat{\mathbb{P}} \mid a(X) \in (a_k, a_{k+1})]$, i.e., the expected calibration error of an input that falls within the atypicality quantile $k$.

**Atypical Examples are Poorly Calibrated:** In Figure 2a, we show the distribution of miscalibration where each bin within the grid contains the intersection of the corresponding confidence and atypicality quantiles. We observe that within the same confidence range, predictions for atypical points have lower accuracies and are more overconfident. In other words, predictions in the Extrapolation or Untrustworthy regions are more miscalibrated than the ones in the typical regions.

In Figure 2b, we split inputs into quantiles according to atypicality and compute the ECE and Accuracy for each group. Results show a monotonic relationship between atypicality and ECE or Accuracy across the three settings. Specifically, we see that predictions for atypical inputs or samples from rare classes are more miscalibrated and have lower accuracy. For samples from rare classes, the model overpredicts the probabilities of the typical class, hence we have overconfidence and low accuracy. Appendix C.3, and §4 present figures and tables for all model and dataset pairs.

### 3.2    Theoretical Analysis: Characterizing Calibration Error with Atypicality

We characterize how calibration error varies with atypicality in a tractable model that is commonly used in machine learning theory [BMWX21a, BMWX21b, ZDKZ22, CLKZ22]. Our theoretical analysis further supports our empirical findings.

**Data Generative Model:** We consider the well-specified logistic model for binary classification with Gaussian data, where $Y \in \{-1, 1\}$ and the $\mathbb{P}(Y = 1|X)$ is defined by the sigmoid function:

$$\mathbb{P}(Y = 1 \mid X) = \sigma(\langle \beta^*, X \rangle), \quad X \sim N(0, I_d).$$

Where $I_d$ denotes the $d$-dimensional identity matrix, $\beta^*$ is the ground truth coefficient vector, $\sigma(x) = 1/(1 + e^{-x})$, and we have $i.i.d.$ observations $\{(x_i, y_i)\}_{i=1}^n$ sampled from the above distribution.

**The Estimator:** We focus on studying the solution produced by minimizing the logistic loss

$$\hat{\beta} = \arg\min_{\beta} \frac{1}{n} \sum_{i=1}^{n} [\log(1 + \exp(\beta^\top x_i)) - y_i \cdot \beta^\top x_i].$$

For $k \in \{-1, 1\}$, $\hat{\mathbb{P}}_k(x)$ is an estimator of $\mathbb{P}(y = k|x)$, with the form $\hat{\mathbb{P}}_k(x) = \frac{1}{e^{-k \cdot \hat{\beta}^\top x} + 1}$.

**Calibration:** We consider all $x$ where $\mathbb{P}_1(x) > 1/2$, as $\mathbb{P}_1(x) \leq 1/2$ can be analyzed similarly by symmetry (see Appendix G). For $u \in (1/2, 1)$, the signed calibration error at a confidence level $u$ is

$$u - \mathbb{P}(Y = 1 \mid \hat{\mathbb{P}}_1(X) = u).$$

We want to show that when $X$ is atypical, i.e., when $a(X) := \|X\|^2/2$ is larger[5], the accuracy $\mathbb{P}(Y = 1 \mid \hat{\mathbb{P}}_1(X) = u)$ would be generally smaller than the confidence $u$ (over-confidence).

**Theorem 3.1.** *Consider the data generative model and the learning setting above. For any $K > 1$, suppose we consider the quantiles of $a(X)$, $a_1, a_2, ..., a_K, a_{K+1}$ such that $\mathbb{P}(a(X) \in (a_k, a_{k+1})) = 1/K$ for $k \in [K]$. We assume $\|\beta^*\| \le c_0$, and $d/n = \kappa$, for some sufficiently small $c_0$. Then, for sufficiently large $n$, for $k = 2, \ldots, K$, we have*

$$\mathbb{E}_{u \sim \hat{\mathbb{P}}_1(X)}[u - \mathbb{P}(Y = 1 \mid \hat{\mathbb{P}}_1(X) = u) \mid a(X) \in (a_k, a_{k+1})] >$$

$$\mathbb{E}_{u \sim \hat{\mathbb{P}}_1(X)}[u - \mathbb{P}(Y = 1 \mid \hat{\mathbb{P}}_1(X) = u) \mid a(X) \in (a_{k-1}, a_k)] \ge 0.$$

That is, the resulting classifier is over-confident, and the level of over-confidence becomes larger when the data is more atypical (with larger $a(X)$). Further, the gap becomes larger for smaller sample sizes $n$. The proof of the theorem is in Appendix G.2 and builds on the results from [BMWX21a, SC19].

## 4 Using Atypicality to Improve Recalibration

Here, we show how atypicality can complement and improve post-hoc calibration. In §2, we observed that predictions for atypical inputs and samples from atypical classes are more overconfident with lower accuracy. We next show that taking input and class atypicality into account improves calibration.

### 4.1 Parametric Recalibration: Different Groups Need Different Temperatures

Temperature scaling (TS), a single parameter variant of Platt Scaling [P+99], is a simple recalibration method that calibrates the model using a single parameter. The predictor is of the form

$$\log \hat{\mathbb{P}}_{\text{TS}}(Y|X) \propto \log \hat{\mathbb{P}}(Y|X)/\tau, \tag{1}$$

where $\hat{\mathbb{P}}(Y|X)$ is the model that takes an input and outputs scores/logits, and $\tau$ is the temperature parameter. In practice, $\tau$ is optimized using a calibration set to minimize a proper scoring rule [GR07, BW19] such as the cross-entropy loss.

To understand the behavior of TS with respect to atypicality, we separately perform TS on points grouped according to the atypicality quantiles. Let us denote the temperature fitted to the quantile covering $a(X) \in (a_{k-1}, a_k]$ by $\tau_{a_k}$. In Appendix Figure 10 we observe an increasing relationship between $a_k$ and $\tau_{a_k}$. Different atypicality groups need different adjustments, and more atypical groups need larger temperatures. *This suggests that being atypicality-aware can improve calibration. While a single temperature value improves average calibration, it may hurt certain groups.*

### 4.2 Atypicality-Aware Recalibration

We showed that predictions are more reliable when the input is typical. However, predictions are less reliable for atypical inputs, and we may need further revision. An analogy can be drawn to decision-making literature where opinions of individuals are combined with geometric averaging weighted by their expertise [FP98, AR89]. Analogously, we propose *Atypicality-Aware Recalibration (AAR)* a method designed to address the reliability issues identified in dealing with atypical inputs:

$$\hat{\mathbb{P}}_{\text{AAR}}(Y|X) = \frac{\hat{\mathbb{P}}(Y|X)^{\psi(a(X))} \exp(S_Y)^{1-\psi(a(X))}}{Z(X)}, \tag{2}$$

where $\psi(a(X))$ is a function of input atypicality, $S_Y$ is a tunable score for class $Y$, $Z(X)$ is the normalization term. Intuitively, when the input is typical, we trust the model confidence; otherwise, we use a score for the given class estimated from the calibration set. Note that this form simplifies to

$$\log \hat{\mathbb{P}}_{\text{AAR}}(Y|X) \propto \phi(a(X)) \log \hat{\mathbb{P}}(Y|X) + S_Y, \tag{3}$$

---

[5]The definition of atypicality follows from the marginal likelihood of the data model: density for the Gaussian with zero mean and identity covariance.

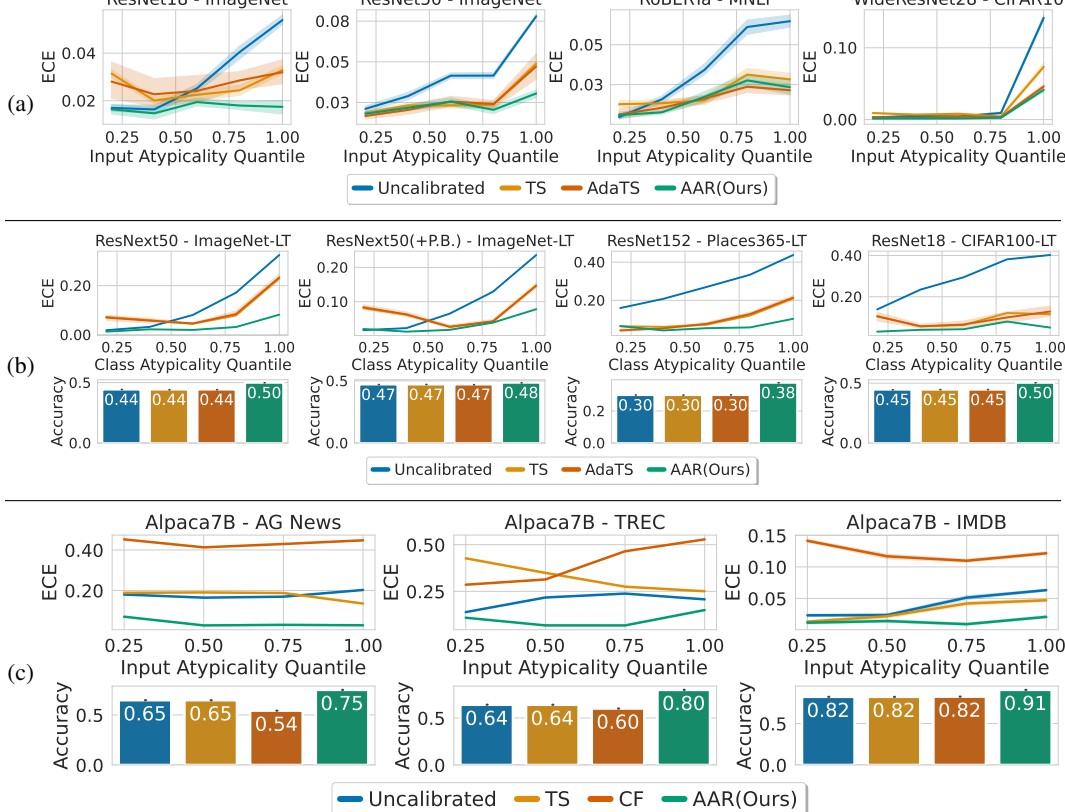

Figure 3: **Post-hoc Recalibration for Classification. (a) Balanced Supervised Classification:** Atypicality-Aware Recalibration improves the calibration of models trained with balanced datasets, across atypicality groups. **(b) Imbalanced Supervised Classification:** Atypicality-Aware Recalibration improves both the calibration across groups and the overall accuracy of models trained with imbalanced datasets. **(c) Classification with LLMs:** Atypicality-Aware Recalibration improves both the calibration across groups and the overall accuracy of LLMs performing classification.

where we subsume $(1 - \psi(a(X))$ into $\phi(a(X))$. We give a simple interpretation of this form: the multiplicative term is an atypicality-dependent temperature, and the additive term is a class-dependent correction where $\exp(S_Y)$ can be considered to induce a correction distribution over classes estimated from the calibration set.

Intuitively, when $\psi(a(X)) = 0$, the output reduces to a fixed distribution over classes that was estimated using the calibration set. This distribution can be seen to induce a prior probability over classes, and $\psi$ controls the tradeoff between this prior and the model's predictive distribution. As the point becomes more typical, this distribution is closer to the model's predictive distribution. In Appendix Figure 11, we show how these values behave with class atypicality. We find that rare classes require larger positive corrections with larger $S_Y$.

**Implementation Details:** Following TS, we minimize the cross-entropy loss on a calibration set. With the temperature-atypicality relationship observed in Figure 10 we choose to instantiate the multiplicative factor as a quadratic function, where $\phi(a(X)) = c_2 a(X)^2 + c_1 a(X) + c_0$ and in total we have $|\{S_1, .., S_{|\mathcal{Y}|}, c_0, c_1, c_2\}| = |\mathcal{Y}| + 3$ interpretable parameters. Once the embeddings and logits are computed, AAR runs on a CPU in under 1 minute for all experimented settings.

Similar to our adaptive interpretation, a concurrent work, Adaptive Temperature Scaling (AdaTS) [JPL+23], uses temperature scaling where the temperature is parameterized by a Variational Autoencoder(VAE) [KW13] and a multi-layer perceptron on top of the VAE embeddings. In the below experiments, we give results with AdaTS as a baseline when applicable.

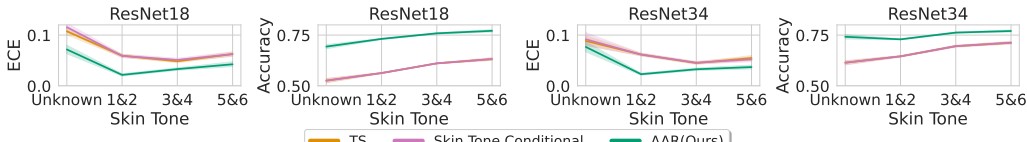

Figure 4: **Improving Group Performance through Atypicality-Awareness.** Here we show that AAR improves the calibration and accuracy of models across different skin tone groups. With AAR, we can improve both the worst group performance and overall performance significantly without using group attributes. TS curve is less visible since it significantly overlaps with Skin Tone Conditional.

For **Balanced Supervised Classification**, in Figure 3a, we observe that being atypicality aware improves recalibration across all groups. We perform comparably to AdaTS, where the temperature function has in the order of millions of parameters, whereas AAR has only $|\mathcal{Y}| + 3$ parameters.

In **Imbalanced Supervised Classification** (Figure 3b), our algorithm not only provides better calibration rates across all classes but also improves overall accuracy. Note that only our method can change accuracy (due to the additive term), and it performs better than other baselines in terms of ECE across all classes. Further, the second column shows using Progressive Balancing [ZWF+21] in training, showing that our post-hoc method can complement methods that modify training procedures.

For **Classification with LLMs**, we add an LLM calibration baseline Content-Free Calibration (CF) [ZWF+21]. We cannot use AdaTS as the embeddings are not fixed in size. In Figure 3c, we see AAR has better calibration and accuracy across the three datasets. Namely, by adjusting the LLM output using the LLM atypicality, we can adjust the probabilities to increase the prediction quality.

### 4.3 Case Study: Fairness through Atypicality-Awareness

Machine learning models reportedly have performance disparity across subgroups [BHN17] due to factors such as varying sample size or noise levels [CJS18]. For instance, skin lesion classifiers can exhibit performance disparity across different skin tones [DVN+22]. Fitzpatrick17k [GHS+21] is a dataset of clinical images with Fitzpatrick skin tone annotations between 1-to-6, where a larger number means darker skin tones, and when annotators do not agree, it is labeled as 'Unknown'. We explore the classification problem with 9 classes indicating the malignancy and the type of skin condition, using a ResNet18/34 pretrained on ImageNet and finetuned on this task (See Appendix F).

When the goal is to improve performance across groups, one can use group annotations and optimize performance within each group [HJKRR18, KGZ19]. Here, we investigate how complementing recalibration with atypicality can improve prediction quality across all groups *without group annotations*. For comparison, we perform 3 recalibration methods: TS, AAR, and Skin-Tone Conditional TS which calibrates the model individually for each skin-tone group with TS. Since the skin-tone conditional calibration uses group attributes, ideally it should act as an oracle.

In Figure 4, we give the Accuracy and ECE analyses where AAR improves performance across all groups. For instance, the worst-group Accuracy (0.69) or ECE (0.072) with AAR is close to the best-group Accuracy (0.63) or ECE (0.062) with the other two methods. Overall, *our findings suggest that Atypicality-Awareness can complement fairness-enforcing methods, and improve performance even when the group annotations are unavailable.* We hypothesize that with AAR, we can perform better than using supervised group attributes since groups may not have sufficient sample size in the calibration set (131, 1950, 1509, 555 samples for Unknown, 1&2, 3&4, and 5&6 respectively), and we can leverage atypicality to offer some mitigation. Further investigating how to leverage atypicality to improve fairness and factors affecting performance disparities is a promising direction for future work [CJS18].

## 5 Improving Prediction Sets with Atypicality

**Conformal Prediction** [SV08, AB21] is a framework that assigns a calibrated prediction set to each instance. The goal is to find a function $\mathcal{C} : \mathcal{X} \rightarrow 2^{\mathcal{Y}}$ that returns a subset of the label space such that $Y \in \mathcal{C}(X)$ with high probability. The framework aims to guarantee *marginal coverage*, i.e., $\mathbb{P}(Y \in \mathcal{C}(X)) \geq 1 - \alpha$, for a choice of $\alpha$. We investigate two conformal calibration methods,

Adaptive Prediction Sets (APS) [RSC20] and Regularized APS (RAPS) [ABMJ20]. Let $\pi(X)$ be the permutation of the label set that sorts $\hat{\mathbb{P}}(Y = c|X)$, i.e. the predicted probabilities for each class $c$ after TS. The prediction sets are produced by the function $\mathcal{C}(x) = \{y : s(x, y) \leq \hat{q}\}$, and these methods fit the threshold $\hat{q}$ for a choice of the scoring function. APS uses the cumulative sum of the predicted probabilities $s(x, y) = \sum_{j=1}^{c} \hat{\mathbb{P}}(Y = j|X)$, where $y = \pi_c(X)$. Intuitively, if the model was perfectly calibrated, we would have expected to have $\hat{q} = 1 - \alpha$. Similarly, RAPS builds on the idea that tail probabilities are noisy and regularizes the number of samples in the prediction set.

Building on our ideas in the previous sections we implement Atypicality-Aware prediction sets, namely *AA-APS* and *AA-RAPS* in the following way: We first group points according to their confidence and atypicality quantiles. A group $G$ here is defined using 4 thresholds, namely $G = x : (l_a^{(G)} < q_a(x) \leq h_a^{(G)}) \wedge (l_c^{(G)} < q_c(x) \leq h_c^{(G)})$ where $q_a(x)$ and $q_c(x)$ denote the atypicality and confidence quantiles for the sample $x$, $l_a^{(G)}$ and $h_a^{(G)}$ denote the atypicality lower and upper bounds for group $G$, and $l_c^{(G)}$ and $h_c^{(G)}$ denote the confidence lower and upper bounds for group $G$. Using a calibration set, these bounds are simply determined by the quantiles of confidence and atypicality statistics. Then, we fit separate thresholds $\hat{q}_G$ for each group's prediction sets with APS or RAPS as subroutines. This allows us to have an adaptive threshold depending only on the atypicality and confidence of predictions.

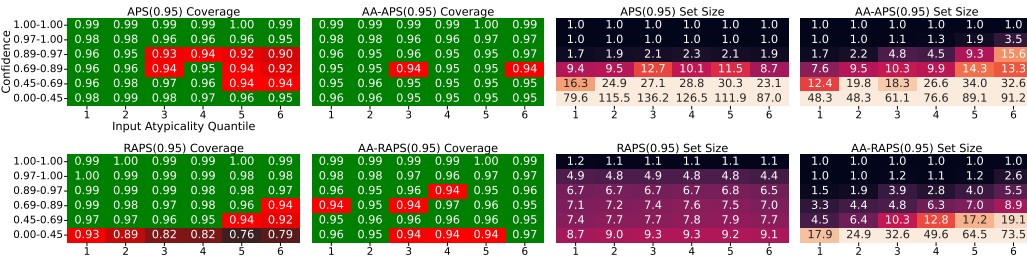

Figure 5: **Improving Conformal Calibration with Atypicality for ResNet50 on ImageNet.** Here we show that atypicality awareness improves conformal calibration performance across different groups. Methods are fitted to satisfy $95\%$ coverage. We observe that APS and RAPS do not satisfy conditional coverage for high atypicality regions or low confidence regions.

In Figure 5, we provide the coverage plots for APS and RAPS in the first and third columns. Even though marginal coverage is satisfied, models do not satisfy conditional coverage for atypical inputs or low-confidence predictions. We observe that being Atypicality-Aware improves coverage across otherwise underperforming groups. Further, AA-APS has lower set sizes on average than APS (15.6 vs 21.3). While RAPS has a lower average set size than AA-RAPS (4.2 vs 9.1) AA-RAPS has smaller set sizes for high-confidence samples, whereas a larger set size for low-confidence samples where the coverage is not met for RAPS. In Appendix D.3, we provide the same analysis for ResNet18,50,152 at different coverage levels along with analyzing the performance in the Confidence and Atypicality dimensions individually. For instance in Figure 8, we observe that RAPS and APS do not satisfy coverage for high atypicality regions, even when averaged across different confidence levels.

## 6 Additional Related Work

**Uncertainty and Atypicality:** [MKvA+21, PBC+20] use density estimation to disentangle epistemic and aleatoric uncertainty. Following this, they show improvements in active learning and OOD detection [LLLS18]. We note that our goal is not this disentanglement (e.g. Untrustworthy quadrant can have both aleatoric or epistemic uncertainty), or Ambiguity could be due to a lack of features or noise. [LLP+20] propose the related notion of distance awareness, and that it leads to better uncertainty quantification. They offer architecture and training modifications whereas we analyze existing models using our framework including imbalanced and LLM settings, and propose simple and post-hoc approaches. 'OOD' [HG16b] or 'anomaly' [HMD18] notions are tied to atypicality, yet our goal is not to make a binary distinction between 'in' or 'out'. We argue that in-distribution samples could also be atypical (e.g. rare groups), and the goal is to perform reliably in the entire spectrum. Other works with an atypicality notion include bounding calibration of groups by the

excess risk [LSH19], miscalibration under distribution shifts [OFR+19], uncertainty in Gaussian Processes [Ras04], forgetting time for rare examples [MGLK22], the poor performance of groups with lower sample sizes [CJS18], energy-based models improving calibration [GWJ+20], relating perplexity to zero-shot classification performance for LLMs [GIB+22], grouping loss and local definitions of miscalibration [PLMV23], the relationship between active learning and atypicality [HDW22], sample size as a factor for subgroup performance disparity [CJS18]. [PBZ21] provide insightful discussion around the nature of softmax confidence, and here we show that its reliability depends on the atypicality of the input. Our new findings include showing that predictions for atypical samples are more miscalibrated and overconfident, and atypicality awareness improves prediction quality. *Overall, while there are other relevant notions in the literature, our distinct goal is to show that post-hoc atypicality estimation and recalibration is a simple yet useful framework to understand and improve uncertainty quantification that complements existing methods.*

**Recalibration and Conformal Prediction:** There is a rich literature on recalibration methods and prediction sets: TS [GPSW17], Platt Scaling [P+99], conformal calibration [SV08, ABMJ20] among many. [LLC+22, RBSC19, BYR+21, BGJ+22] make a relevant observation, showing that the coverage of conformal prediction is not equal across all groups. They propose group conformal calibration, which requires group labels whereas our proposal is unsupervised and does not depend on any attribute information. Concurrent work [JPL+23] explores AdaTS, where they train a separate VAE and MLP to produce an adaptive temperature. However, our parameterization of temperature has 3 parameters and is interpretable.

## 7 Conclusion

Atypicality offers a simple yet flexible framework to better understand and improve model reliability and uncertainty. We propose that pretrained models should be released not only with confidence but also with an atypicality estimator. While there are other relevant notions in the literature, our main goal is to show that atypicality can provide a unifying perspective to discuss uncertainty, understand individual data points, and improve fairness. Here we focus on classification problems; it would be interesting to extend atypicality to regression and generation settings. Furthermore, we would like to extend the theoretical analysis to more general settings, as our empirical results demonstrate that the observed phenomena hold more broadly.

## Acknowledgments

We would like to thank Adarsh Jeewajee, Bryan He, Edward Chen, Federico Bianchi, Kyle Swanson, Natalie Dullerud, Ransalu Senanayake, Sabri Eyuboglu, Shirley Wu, Weixin Liang, Xuechen Li, Yongchan Kwon, Yu Sun, and Zach Izzo for their comments and suggestions on the manuscript. Linjun Zhang's research is partially supported by NSF DMS-2015378. Carlos Ernesto Guestrin is a Chan Zuckerberg Biohub – San Francisco Investigator.

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

# A  Atypicality Estimation

## A.1  Input Atypicality Estimation

To estimate input atypicality, we use two ways to estimate the likelihood of a point under the training distribution. First, we give methods for the discriminative models.

**Fitting individual Gaussians to class conditionals**  Here, we follow a similar approach to [MKvA$^+$21]. Namely, we model the class conditionals with a Gaussian distribution, where the covariance matrix is tied across classes:

$$\hat{\mathbb{P}}(X|Y = y) \sim N(X; \mu_y, \Sigma) \tag{4}$$

We fit the parameters $\mu_y$ and $\Sigma$ with maximum likelihood estimation. The reason to tie the covariance matrix is due to the number of samples required to fit the density. Namely, for a $d$-dimensional problem, the total number of parameters to fit individual matrices becomes $O(yd^2)$, which results in low-quality estimates. Then, the atypicality becomes

$$a_X(x) = -\max_{y \in \mathcal{Y}} \log \hat{\mathbb{P}}(X = x|Y = y) \tag{5}$$

**Computing distance with k-Nearest Neighbors  k-Nearest Neighbors:**  Similarly, we can use the nearest neighbor distance. Concretely, we use the nearest neighbor distance, $a_X(x) = d_{\min}(x, \mathcal{D}_{\text{train}}) = \min_{x' \in \mathcal{D}_{\text{train}}} |x' - x|$, as the atypicality metric. Alternatively, we can use different notions such as the average of k-nearest neighbors, or the distance to the kth neighbor. Below, we report the results by using the average distance to 5-nearest neighbors.

**Fitting the estimators**  For all of the atypicality estimators, we fit the estimators using samples from the training sets and make inferences for the calibration and test sets. For instance, we use the training split of ImageNet to fit the corresponding density estimator and compute the atypicality for the samples from the validation/test split of ImageNet. All of our results using atypicality are reported for the test splits of the below datasets.

**Atypicality Estimation with LLMs:**  For language models we simply compute the negative log-likelihood of each prompt as the atypicality metric: $a_X(x) = -\log \hat{\mathbb{P}}_{\text{LLM}}(x)$. To define confidence, we use the logits of the language model, conditioned on the prompt. We use the logit of the first token of each class label and compute the predicted probabilities by applying softmax to the logits of each class.

## A.2  Class Atypicality

To estimate class atypicality, we simply count the fraction of examples from a particular label in the training dataset. Let us have a training dataset $\mathcal{D}_{\text{train}} = \{(x_1, y_1), (x_2, y_2), \ldots, (x_N, y_N)\}$. Then, we estimate class atypicality with

$$a_Y(y) = -\log \frac{\sum_{i \in [N]} \mathbf{1}[y_i = y]}{N}. \tag{6}$$

## A.3  Atypicality and Confidence

Are atypicality and confidence equally informative? Beyond the data perspective given in Figure 1, here we provide quantitative results to demonstrate the difference. In Figure 6, we give a grid plot where the x-axis indicates the typicality quantile of a point, and the y-axis indicates the confidence of a point. The coloring on the left indicates the accuracy within a bin split according to accuracy, and the right has the difference between average confidence and accuracy. Observe that for a specific confidence interval, larger values of typicality mean better quality probabilistic estimates and larger atypicality means more miscalibration.

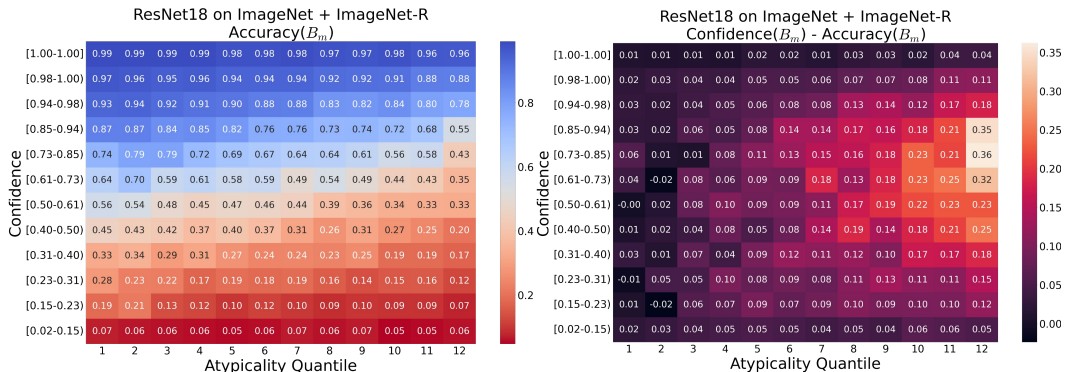

Figure 6: **Input Atypicality and Confidence.** Here, the x-axis reflects the input atypicality quantile, and the y-axis indicates confidence. The coloring for the figure on the left indicates the accuracy within a bin, and the figure on the right has the difference between confidence and accuracy within a bin. We observe that even within the same confidence range, atypical examples tend to be more miscalibrated and overconfident compared to typical examples.

## B  Experimental Details

### B.1  Balanced Supervised Classification

#### B.1.1  Datasets

Below is a full list of datasets for balanced classification:

1. **ImageNet** [DDS$^+$09] from Torchvision [MR10] is an object recognition dataset with 1000 classes. We use the ImageNet-1k version.

2. **CIFAR10/100** [Kri09] from Torchvision [MR10] are object recognition datasets with 10/100 classes.

3. **MNLI** [WNB18] from Huggingface Datasets [LVdMJ$^+$21] is a natural language inference dataset with 3 classes, indicating entailment, neutral, and contradiction outcomes.

#### B.1.2  Models

Most of the models are public models, e.g., obtained from the Transformers Library [WDS$^+$20] or Torchvision [MR10]. Below we give the full model details and how one can access them:

1. **RoBERTa**(`HuggingFace roberta-large-mnli`) trained on the MNLI dataset. One can use the id given here on HuggingFace to download the model.

2. **ResNet18, ResNet50, ResNet152** from (`Torchvision` [MR10]) trained on ImageNet.

3. **WideResNet28** trained on CIFAR10,100 obtained from [MKS$^+$20].

For all BERT [DCLT19] style models we use the `[CLS]` token embeddings in the final layer to perform classification. For all vision models, we use the penultimate layer embeddings to fit the density estimators and perform the analyses. In the experiments, we randomly split the test sets into two equal halves to have a calibration split and a test split, and repeat the experiments over 10 random seeds.

## B.2 Imbalanced Supervised Classification

### B.2.1 Datasets

All of our imbalanced classification datasets are previous benchmarks obtained from the GitHub repository[6] of [ZCLJ21] with corresponding training, validation, and test splits. All of these datasets have an exponential class imbalance.

1. **ImageNet-LT** is the long-tailed variant of ImageNet with 1000 classes.
2. **CIFAR10/100-LT** is the long-tailed variant of CIFAR10/100 with 10/100 classes.
3. **Places365-LT** is the long-tailed variant of Places365 [ZKL+16] with 365 classes.

### B.2.2 Models

Similarly, most of these models are obtained from [ZCLJ21].

1. **ResNeXt50** trained on ImageNet-LT with and without Progressive Balancing, which is a strategy to address class imbalance during training.
2. **ResNet152** trained on Places365-LT
3. **ResNet18** trained on CIFAR100-LT trained by us. This model is pretrained on ImageNet and finetuned on CIFAR100-LT.

We use the validation splits of these datasets as the calibration set, and report the results on the test set.

## B.3 Classification with LLMs

### B.3.1 Model

We use Alpaca-7B [TGZ+23] in a zero-shot setting, where we simply prompt the model with the classification question. We use the prompting strategy from Content-Free Calibration [ZWF+21].

Below, we show examples of each dataset and prompt.

### B.3.2 Datasets

**IMDB** is a binary classification dataset of movie reviews, where the goal is to classify the sentiment in a review. The example prompt has the form 'The following review was written for a movie: [Review].\n What is the sentiment, Positive or Negative? Answer: '. Below is an example:

> The following review was written for a movie: I and a friend rented this movie. We both found the movie soundtrack and production techniques to be lagging. The movie's plot appeared to drag on throughout with little surprise in the ending. We both agreed that the movie could have been compressed into roughly an hour giving it more suspense and moving plot.
> What is the sentiment, Positive or Negative? Answer:

where the correct answer should be 'Negative'. We filter out the examples that exceed the context length limit of Alpaca7B (512). We noticed that the 'validation' split of IMDB leads to significantly worse calibration compared to splitting the test set. Thus, for all experiments, we use the test split of IMDB and split it into 2 sets (instead of using the validation split as a calibration set as in the other two datasets).

**TREC** is a 6-class question classification dataset where the goal is to predict whether a question will have an answer that is an 'Abbreviation', 'Entity', 'Description', 'Human', 'Location', or a 'Number'. We format the prompts with 'Classify the questions based on their Answer Type. Potential Answer Types are: Number, Location, Person, Description, Entity, or Abbreviation.\n\nQuestion: [question]\n\nAnswer Type: '. Below is an example prompt:

---

[6] https://github.com/dvlab-research/MiSLAS

> Classify the questions based on their Answer Type. Potential Answer Types are: Number, Location, Person, Description, Entity, or Abbreviation.
>
> Question: What county is Modesto , California in ?
> Answer Type:

where the correct answer should be 'Location'.

**AG News** is a news classification dataset. The goal is to classify a given news into 4 potential classes: 'World', 'Sports', 'Business', or 'Science and Technology'. We format the prompts with 'Classify the news articles into the categories of World, Sports, Business, and Technology.\n\n Article: [article]\n\nAnswer: '. Below is an example prompt:

> Classify the news articles into the categories of World, Sports, Business, and Technology.
>
> Article: Wall St. Bears Claw Back Into the Black (Reuters) Reuters - Short-sellers, Wall Street's dwindling band of ultra-cynics, are seeing green again.
> Answer:

where the correct answer should be 'Business'.

Furthermore, we also use [ZWF$^+$21] as another calibration baseline. Following their paper, we use N/A, [MASK], and the empty string as content-free. Concretely, we follow their paper to first obtain the average predicted probabilities for each label token for the content-free input, denoted by $p_{cf}$. We then let

$$W = \text{diag}(p_{cf})^{-1}.$$

When making test-time predictions, we compute $\text{Softmax}(W^T p)$ as the new predicted probabilities. In our experiments, we observe that it does not perform as well in this setting, as was previously suggested by [Kum22].

## C   Calibration

We run all our experiments with 10 different random seeds, where the seeds are $\{0, 1, 2, \ldots, 9\}$. Randomness is over fitting the atypicality estimators, and calibration-test splits (we use the same splits with the recalibration experiments for the sake of consistency).

### C.1   Expected Calibration Error

To compute ECE, we generate $\mathbb{B} = \{B_1, B_2, ..., B_M\}$, $M$ equally-spaced bins where samples are sorted and grouped according to their confidence. $B_m$ here denotes the set of the data indices where the confidence of the prediction for the sample falls into the interval $(\frac{m-1}{M}, \frac{m}{M}]$. We compute ECE by

$$\text{ECE}[\hat{\mathbb{P}}] = \sum_{m=1}^{M} \frac{|B_m|}{N} |\text{acc}(B_m) - \text{conf}(B_m)|, \tag{7}$$

where $\text{acc}(B_m) = \frac{1}{|B_m|} \sum_{i=1}^{|B_m|} \mathbf{1}[\hat{y}_i = y_i]$ is the accuracy for the bin $m$, and $\text{conf}(B_m) = \frac{1}{|B_m|} \sum_{i=1}^{|B_m|} \hat{\mathbb{P}}(Y = \hat{y}_i | X = x_i)$ gives the average confidence within the bin. $|B_m|$ is the size of the bin $m$, $N$ is the total number of samples, and $\mathbf{1}[\cdot]$ is the indicator function.

Throughout our experiments, we let the number of bins $|\mathbb{B}| = 10$ by default when computing ECE.

Similarly, below we report results with RMSCE (Root Mean Squared Error) [HMD18] as another calibration metric, which is formulated as the following:

$$\text{RMSCE}[\hat{\mathbb{P}}] = \sqrt{\sum_{m=1}^{M} \frac{|B_m|}{N} (\text{acc}(B_m) - \text{conf}(B_m))^2} \tag{8}$$

## C.2 ECE and Atypicality Results with Different Atypicality Metrics

We further experiment with different atypicality metrics, such as the average distance to the 5-nearest neighbors (Figure 7). We broadly observe that while there are slight differences in the quantitative results between different atypicality metrics, the qualitative phenomena remain intact. In Tables 1, 2, and 3 we give all the results in the tabular form.

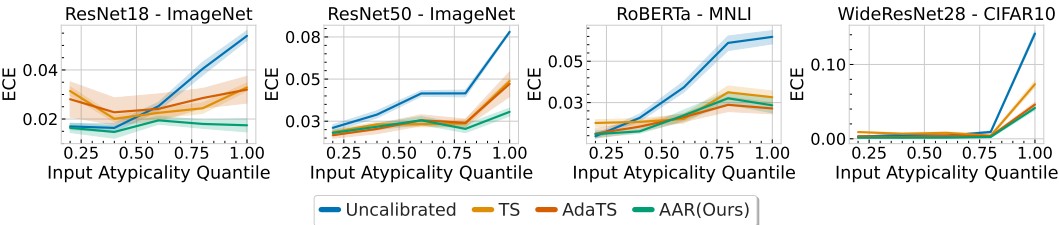

Figure 7: **Atypicality with 5-nearest neighbors and Uncertainty.** Here, we report the results of the same experiments as Figure 3 with the average of the distance to the 10-nearest neighbors as the atypicality metric. See Tables 1 for the results in tabular format.

## C.3 Results in the Tabular Format

Table 1: **Recalibration Results for Balanced Classification.** For each dataset and atypicality quantile, the best results are marked in bold. We provide the standard errors next to the means over 10 random seeds.

| Atypicality | | | | Uncalibrated ECE | Uncalibrated RMSE | TS ECE | TS RMSE | AdaTS ECE | AdaTS RMSE | Atypicality-Aware ECE | Atypicality-Aware RMSE |
|---|---|---|---|---|---|---|---|---|---|---|---|
| GMM | ResNet152 | ImageNet | 0.2 | 0.029 ± 0.001 | 0.077 ± 0.001 | 0.014 ± 0.001 | 0.066 ± 0.002 | 0.014 ± 0.001 | 0.064 ± 0.002 | 0.016 ± 0.001 | 0.065 ± 0.001 |
| | | | 0.4 | 0.039 ± 0.001 | 0.086 ± 0.001 | 0.020 ± 0.001 | 0.074 ± 0.001 | 0.020 ± 0.002 | 0.075 ± 0.002 | 0.019 ± 0.001 | 0.070 ± 0.001 |
| | | | 0.6 | 0.050 ± 0.001 | 0.097 ± 0.001 | 0.026 ± 0.001 | 0.079 ± 0.002 | 0.025 ± 0.002 | 0.081 ± 0.002 | 0.026 ± 0.000 | 0.078 ± 0.001 |
| | | | 0.8 | 0.062 ± 0.001 | 0.106 ± 0.001 | 0.024 ± 0.001 | 0.076 ± 0.001 | 0.023 ± 0.002 | 0.078 ± 0.002 | 0.024 ± 0.001 | 0.077 ± 0.001 |
| | | | 1.0 | 0.084 ± 0.001 | 0.123 ± 0.001 | 0.037 ± 0.001 | 0.088 ± 0.001 | 0.033 ± 0.004 | 0.085 ± 0.003 | 0.026 ± 0.001 | 0.080 ± 0.001 |
| | ResNet18 | ImageNet | 0.2 | 0.017 ± 0.001 | 0.074 ± 0.001 | 0.031 ± 0.001 | 0.084 ± 0.001 | 0.028 ± 0.004 | 0.081 ± 0.004 | 0.016 ± 0.001 | 0.069 ± 0.001 |
| | | | 0.4 | 0.016 ± 0.001 | 0.074 ± 0.001 | 0.020 ± 0.001 | 0.079 ± 0.001 | 0.023 ± 0.003 | 0.079 ± 0.003 | 0.015 ± 0.001 | 0.072 ± 0.002 |
| | | | 0.6 | 0.025 ± 0.001 | 0.080 ± 0.001 | 0.023 ± 0.001 | 0.077 ± 0.001 | 0.024 ± 0.001 | 0.080 ± 0.002 | 0.019 ± 0.001 | 0.074 ± 0.001 |
| | | | 0.8 | 0.040 ± 0.001 | 0.092 ± 0.001 | 0.024 ± 0.001 | 0.078 ± 0.001 | 0.029 ± 0.002 | 0.082 ± 0.003 | 0.018 ± 0.001 | 0.074 ± 0.001 |
| | | | 1.0 | 0.054 ± 0.001 | 0.100 ± 0.001 | 0.033 ± 0.001 | 0.085 ± 0.001 | 0.032 ± 0.003 | 0.083 ± 0.003 | 0.017 ± 0.002 | 0.073 ± 0.002 |
| | ResNet50 | ImageNet | 0.2 | 0.021 ± 0.001 | 0.069 ± 0.002 | 0.018 ± 0.001 | 0.071 ± 0.001 | 0.017 ± 0.001 | 0.068 ± 0.002 | 0.018 ± 0.001 | 0.067 ± 0.002 |
| | | | 0.4 | 0.029 ± 0.001 | 0.079 ± 0.001 | 0.023 ± 0.001 | 0.077 ± 0.001 | 0.020 ± 0.002 | 0.074 ± 0.001 | 0.022 ± 0.001 | 0.074 ± 0.001 |
| | | | 0.6 | 0.041 ± 0.001 | 0.091 ± 0.001 | 0.023 ± 0.001 | 0.078 ± 0.001 | 0.026 ± 0.002 | 0.079 ± 0.002 | 0.026 ± 0.001 | 0.078 ± 0.001 |
| | | | 0.8 | 0.042 ± 0.001 | 0.092 ± 0.001 | 0.024 ± 0.001 | 0.078 ± 0.001 | 0.024 ± 0.001 | 0.080 ± 0.002 | 0.020 ± 0.001 | 0.075 ± 0.001 |
| | | | 1.0 | 0.078 ± 0.000 | 0.118 ± 0.000 | 0.049 ± 0.001 | 0.095 ± 0.001 | 0.047 ± 0.004 | 0.096 ± 0.003 | 0.031 ± 0.001 | 0.082 ± 0.001 |
| | RoBERTa | MNLI | 0.2 | 0.005 ± 0.001 | 0.062 ± 0.004 | 0.013 ± 0.001 | 0.077 ± 0.002 | 0.006 ± 0.002 | 0.063 ± 0.005 | 0.006 ± 0.001 | 0.063 ± 0.002 |
| | | | 0.4 | 0.016 ± 0.001 | 0.091 ± 0.004 | 0.013 ± 0.001 | 0.083 ± 0.002 | 0.010 ± 0.002 | 0.075 ± 0.005 | 0.008 ± 0.001 | 0.072 ± 0.002 |
| | | | 0.6 | 0.034 ± 0.002 | 0.116 ± 0.002 | 0.015 ± 0.001 | 0.092 ± 0.001 | 0.016 ± 0.001 | 0.093 ± 0.005 | 0.017 ± 0.002 | 0.090 ± 0.004 |
| | | | 0.8 | 0.061 ± 0.002 | 0.153 ± 0.003 | 0.031 ± 0.002 | 0.113 ± 0.004 | 0.024 ± 0.002 | 0.104 ± 0.004 | 0.028 ± 0.002 | 0.114 ± 0.004 |
| | | | 1.0 | 0.065 ± 0.002 | 0.156 ± 0.002 | 0.028 ± 0.002 | 0.112 ± 0.003 | 0.021 ± 0.002 | 0.106 ± 0.004 | 0.023 ± 0.003 | 0.107 ± 0.004 |
| | WideResNet28 | CIFAR10 | 0.2 | 0.003 ± 0.000 | 0.041 ± 0.001 | 0.009 ± 0.000 | 0.057 ± 0.002 | 0.003 ± 0.000 | 0.051 ± 0.002 | 0.001 ± 0.000 | 0.041 ± 0.001 |
| | | | 0.4 | 0.005 ± 0.001 | 0.051 ± 0.002 | 0.007 ± 0.001 | 0.055 ± 0.001 | 0.002 ± 0.001 | 0.047 ± 0.002 | 0.002 ± 0.000 | 0.048 ± 0.002 |
| | | | 0.6 | 0.004 ± 0.001 | 0.050 ± 0.003 | 0.008 ± 0.001 | 0.058 ± 0.001 | 0.004 ± 0.001 | 0.049 ± 0.004 | 0.002 ± 0.000 | 0.044 ± 0.002 |
| | | | 0.8 | 0.009 ± 0.001 | 0.066 ± 0.003 | 0.004 ± 0.001 | 0.056 ± 0.002 | 0.003 ± 0.001 | 0.058 ± 0.003 | 0.002 ± 0.001 | 0.055 ± 0.002 |
| | | | 1.0 | 0.142 ± 0.002 | 0.240 ± 0.002 | 0.073 ± 0.002 | 0.168 ± 0.003 | 0.046 ± 0.002 | 0.129 ± 0.004 | 0.041 ± 0.002 | 0.123 ± 0.002 |
| | WideResNet28 | CIFAR100 | 0.2 | 0.007 ± 0.001 | 0.059 ± 0.003 | 0.018 ± 0.001 | 0.084 ± 0.001 | 0.021 ± 0.003 | 0.089 ± 0.005 | 0.021 ± 0.001 | 0.094 ± 0.002 |
| | | | 0.4 | 0.029 ± 0.001 | 0.110 ± 0.002 | 0.005 ± 0.001 | 0.070 ± 0.003 | 0.008 ± 0.002 | 0.073 ± 0.003 | 0.005 ± 0.001 | 0.078 ± 0.003 |
| | | | 0.6 | 0.101 ± 0.002 | 0.201 ± 0.002 | 0.052 ± 0.001 | 0.136 ± 0.002 | 0.044 ± 0.004 | 0.124 ± 0.007 | 0.049 ± 0.001 | 0.132 ± 0.003 |
| | | | 0.8 | 0.257 ± 0.004 | 0.297 ± 0.002 | 0.106 ± 0.003 | 0.191 ± 0.003 | 0.101 ± 0.003 | 0.187 ± 0.002 | 0.105 ± 0.003 | 0.190 ± 0.003 |
| | | | 1.0 | 0.371 ± 0.004 | 0.351 ± 0.002 | 0.105 ± 0.004 | 0.200 ± 0.003 | 0.101 ± 0.004 | 0.199 ± 0.003 | 0.109 ± 0.005 | 0.207 ± 0.005 |
| KNN | ResNet152 | ImageNet | 0.2 | 0.035 ± 0.001 | 0.085 ± 0.001 | 0.014 ± 0.001 | 0.062 ± 0.002 | 0.016 ± 0.002 | 0.068 ± 0.001 | 0.011 ± 0.001 | 0.061 ± 0.001 |
| | | | 0.4 | 0.039 ± 0.001 | 0.085 ± 0.001 | 0.015 ± 0.001 | 0.067 ± 0.001 | 0.018 ± 0.001 | 0.069 ± 0.002 | 0.017 ± 0.001 | 0.071 ± 0.001 |
| | | | 0.6 | 0.035 ± 0.001 | 0.083 ± 0.001 | 0.021 ± 0.001 | 0.076 ± 0.002 | 0.024 ± 0.003 | 0.079 ± 0.004 | 0.019 ± 0.001 | 0.072 ± 0.001 |
| | | | 0.8 | 0.057 ± 0.002 | 0.102 ± 0.001 | 0.031 ± 0.001 | 0.084 ± 0.001 | 0.033 ± 0.003 | 0.085 ± 0.003 | 0.032 ± 0.001 | 0.082 ± 0.001 |
| | | | 1.0 | 0.099 ± 0.001 | 0.129 ± 0.001 | 0.036 ± 0.001 | 0.090 ± 0.001 | 0.042 ± 0.006 | 0.093 ± 0.004 | 0.037 ± 0.001 | 0.091 ± 0.001 |
| | ResNet18 | ImageNet | 0.2 | 0.025 ± 0.001 | 0.077 ± 0.002 | 0.020 ± 0.001 | 0.074 ± 0.001 | 0.037 ± 0.021 | 0.115 ± 0.042 | 0.019 ± 0.001 | 0.072 ± 0.001 |
| | | | 0.4 | 0.017 ± 0.001 | 0.072 ± 0.001 | 0.021 ± 0.001 | 0.076 ± 0.001 | 0.045 ± 0.025 | 0.120 ± 0.046 | 0.019 ± 0.001 | 0.074 ± 0.001 |
| | | | 0.6 | 0.021 ± 0.001 | 0.074 ± 0.001 | 0.027 ± 0.001 | 0.083 ± 0.001 | 0.052 ± 0.029 | 0.130 ± 0.050 | 0.020 ± 0.001 | 0.074 ± 0.002 |
| | | | 0.8 | 0.033 ± 0.002 | 0.085 ± 0.001 | 0.022 ± 0.001 | 0.079 ± 0.001 | 0.061 ± 0.038 | 0.136 ± 0.058 | 0.021 ± 0.001 | 0.077 ± 0.001 |
| | | | 1.0 | 0.054 ± 0.001 | 0.102 ± 0.001 | 0.030 ± 0.001 | 0.084 ± 0.001 | 0.079 ± 0.045 | 0.152 ± 0.064 | 0.027 ± 0.001 | 0.081 ± 0.002 |
| | ResNet50 | ImageNet | 0.2 | 0.029 ± 0.001 | 0.078 ± 0.001 | 0.016 ± 0.001 | 0.066 ± 0.001 | 0.015 ± 0.002 | 0.068 ± 0.002 | 0.016 ± 0.001 | 0.067 ± 0.001 |
| | | | 0.4 | 0.029 ± 0.001 | 0.078 ± 0.001 | 0.018 ± 0.001 | 0.072 ± 0.001 | 0.019 ± 0.001 | 0.074 ± 0.002 | 0.017 ± 0.001 | 0.070 ± 0.002 |
| | | | 0.6 | 0.029 ± 0.001 | 0.078 ± 0.001 | 0.021 ± 0.001 | 0.077 ± 0.001 | 0.020 ± 0.002 | 0.077 ± 0.002 | 0.020 ± 0.001 | 0.075 ± 0.001 |
| | | | 0.8 | 0.046 ± 0.001 | 0.096 ± 0.001 | 0.028 ± 0.001 | 0.079 ± 0.001 | 0.026 ± 0.001 | 0.079 ± 0.001 | 0.031 ± 0.001 | 0.082 ± 0.001 |
| | | | 1.0 | 0.076 ± 0.001 | 0.116 ± 0.001 | 0.039 ± 0.002 | 0.092 ± 0.001 | 0.040 ± 0.004 | 0.091 ± 0.003 | 0.038 ± 0.002 | 0.090 ± 0.001 |
| | RoBERTa | MNLI | 0.2 | 0.003 ± 0.000 | 0.056 ± 0.003 | 0.011 ± 0.001 | 0.079 ± 0.002 | 0.011 ± 0.002 | 0.067 ± 0.005 | 0.003 ± 0.001 | 0.061 ± 0.003 |
| | | | 0.4 | 0.014 ± 0.001 | 0.080 ± 0.003 | 0.015 ± 0.002 | 0.091 ± 0.003 | 0.016 ± 0.004 | 0.092 ± 0.007 | 0.008 ± 0.001 | 0.074 ± 0.002 |
| | | | 0.6 | 0.029 ± 0.002 | 0.111 ± 0.003 | 0.014 ± 0.001 | 0.077 ± 0.002 | 0.023 ± 0.005 | 0.095 ± 0.008 | 0.015 ± 0.001 | 0.081 ± 0.005 |
| | | | 0.8 | 0.067 ± 0.002 | 0.156 ± 0.003 | 0.034 ± 0.002 | 0.108 ± 0.004 | 0.030 ± 0.003 | 0.107 ± 0.004 | 0.026 ± 0.003 | 0.101 ± 0.005 |
| | | | 1.0 | 0.064 ± 0.002 | 0.153 ± 0.003 | 0.027 ± 0.002 | 0.110 ± 0.003 | 0.033 ± 0.007 | 0.119 ± 0.009 | 0.026 ± 0.003 | 0.107 ± 0.005 |
| | WideResNet28 | CIFAR10 | 0.2 | 0.001 ± 0.000 | 0.031 ± 0.003 | 0.011 ± 0.000 | 0.061 ± 0.001 | 0.005 ± 0.001 | 0.049 ± 0.004 | 0.003 ± 0.000 | 0.037 ± 0.001 |
| | | | 0.4 | 0.000 ± 0.000 | 0.010 ± 0.005 | 0.012 ± 0.000 | 0.062 ± 0.001 | 0.007 ± 0.001 | 0.054 ± 0.004 | 0.004 ± 0.000 | 0.037 ± 0.001 |
| | | | 0.6 | 0.005 ± 0.000 | 0.049 ± 0.002 | 0.007 ± 0.001 | 0.055 ± 0.002 | 0.003 ± 0.001 | 0.046 ± 0.004 | 0.001 ± 0.000 | 0.048 ± 0.002 |
| | | | 0.8 | 0.009 ± 0.001 | 0.061 ± 0.001 | 0.004 ± 0.001 | 0.058 ± 0.001 | 0.004 ± 0.001 | 0.062 ± 0.003 | 0.003 ± 0.000 | 0.052 ± 0.002 |
| | | | 1.0 | 0.148 ± 0.002 | 0.241 ± 0.002 | 0.079 ± 0.003 | 0.170 ± 0.003 | 0.053 ± 0.004 | 0.136 ± 0.005 | 0.039 ± 0.003 | 0.121 ± 0.003 |
| | WideResNet28 | CIFAR100 | 0.2 | 0.006 ± 0.000 | 0.054 ± 0.002 | 0.020 ± 0.001 | 0.089 ± 0.001 | 0.021 ± 0.002 | 0.089 ± 0.003 | 0.014 ± 0.001 | 0.083 ± 0.001 |
| | | | 0.4 | 0.022 ± 0.001 | 0.092 ± 0.002 | 0.004 ± 0.001 | 0.070 ± 0.002 | 0.006 ± 0.001 | 0.067 ± 0.004 | 0.005 ± 0.001 | 0.076 ± 0.002 |
| | | | 0.6 | 0.090 ± 0.001 | 0.183 ± 0.002 | 0.054 ± 0.001 | 0.140 ± 0.002 | 0.049 ± 0.003 | 0.132 ± 0.004 | 0.041 ± 0.002 | 0.120 ± 0.002 |
| | | | 0.8 | 0.266 ± 0.005 | 0.295 ± 0.003 | 0.124 ± 0.004 | 0.205 ± 0.003 | 0.118 ± 0.004 | 0.202 ± 0.003 | 0.091 ± 0.003 | 0.185 ± 0.003 |
| | | | 1.0 | 0.380 ± 0.003 | 0.356 ± 0.002 | 0.086 ± 0.003 | 0.187 ± 0.004 | 0.087 ± 0.004 | 0.185 ± 0.005 | 0.117 ± 0.005 | 0.216 ± 0.004 |

Table 2: **Recalibration Results for Imbalanced Classification.** For each dataset and atypicality quantile, the best results are marked in bold. We provide the standard errors next to the means over 10 random seeds.

| Atypicality | | | | Uncalibrated ECE | RMSE | Accuracy | TS ECE | RMSE | Accuracy | AdaTS ECE | RMSE | Accuracy | Atypicality-Aware ECE | RMSE | Accuracy |
|---|---|---|---|---|---|---|---|---|---|---|---|---|---|---|---|
| GMM | ResNet152 | Places365-LT | 0.2 | 0.159 ± 0.000 | 0.144 ± 0.000 | 0.492 ± 0.000 | 0.064 ± 0.000 | 0.099 ± 0.000 | 0.492 ± 0.000 | 0.044 ± 0.004 | 0.088 ± 0.002 | 0.492 ± 0.000 | 0.065 ± 0.000 | 0.101 ± 0.000 | 0.412 ± 0.000 |
| | | | 0.4 | 0.208 ± 0.000 | 0.166 ± 0.000 | 0.387 ± 0.000 | 0.061 ± 0.000 | 0.095 ± 0.000 | 0.387 ± 0.000 | 0.053 ± 0.001 | 0.099 ± 0.001 | 0.387 ± 0.000 | 0.042 ± 0.000 | 0.090 ± 0.000 | 0.415 ± 0.000 |
| | | | 0.6 | 0.270 ± 0.000 | 0.186 ± 0.000 | 0.317 ± 0.000 | 0.073 ± 0.000 | 0.111 ± 0.000 | 0.317 ± 0.000 | 0.076 ± 0.005 | 0.111 ± 0.003 | 0.317 ± 0.000 | 0.054 ± 0.000 | 0.097 ± 0.000 | 0.398 ± 0.000 |
| | | | 0.8 | 0.334 ± 0.000 | 0.214 ± 0.000 | 0.218 ± 0.000 | 0.120 ± 0.000 | 0.150 ± 0.000 | 0.218 ± 0.000 | 0.127 ± 0.006 | 0.152 ± 0.002 | 0.218 ± 0.000 | 0.058 ± 0.000 | 0.103 ± 0.000 | 0.371 ± 0.000 |
| | | | 1.0 | 0.437 ± 0.000 | 0.243 ± 0.000 | 0.081 ± 0.000 | 0.211 ± 0.000 | 0.185 ± 0.000 | 0.081 ± 0.000 | 0.212 ± 0.007 | 0.187 ± 0.002 | 0.081 ± 0.000 | 0.103 ± 0.000 | 0.137 ± 0.000 | 0.283 ± 0.000 |
| | ResNet18 | CIFAR10-LT | 0.2 | 0.017 ± 0.000 | 0.077 ± 0.000 | 0.927 ± 0.000 | 0.060 ± 0.000 | 0.164 ± 0.000 | 0.927 ± 0.000 | 0.084 ± 0.008 | 0.190 ± 0.008 | 0.927 ± 0.000 | 0.030 ± 0.000 | 0.104 ± 0.000 | 0.874 ± 0.000 |
| | | | 0.4 | 0.054 ± 0.000 | 0.145 ± 0.000 | 0.825 ± 0.000 | 0.082 ± 0.000 | 0.182 ± 0.000 | 0.825 ± 0.000 | 0.102 ± 0.008 | 0.201 ± 0.006 | 0.825 ± 0.000 | 0.027 ± 0.000 | 0.104 ± 0.000 | 0.741 ± 0.000 |
| | | | 0.6 | 0.152 ± 0.000 | 0.236 ± 0.000 | 0.672 ± 0.000 | 0.049 ± 0.000 | 0.133 ± 0.000 | 0.672 ± 0.000 | 0.069 ± 0.005 | 0.174 ± 0.007 | 0.672 ± 0.000 | 0.044 ± 0.000 | 0.127 ± 0.000 | 0.770 ± 0.000 |
| | | | 0.8 | 0.098 ± 0.000 | 0.188 ± 0.000 | 0.779 ± 0.000 | 0.022 ± 0.000 | 0.107 ± 0.000 | 0.779 ± 0.000 | 0.032 ± 0.005 | 0.113 ± 0.007 | 0.779 ± 0.000 | 0.034 ± 0.000 | 0.104 ± 0.000 | 0.810 ± 0.000 |
| | | | 1.0 | 0.244 ± 0.000 | 0.286 ± 0.000 | 0.584 ± 0.000 | 0.128 ± 0.000 | 0.206 ± 0.000 | 0.584 ± 0.000 | 0.157 ± 0.011 | 0.248 ± 0.009 | 0.584 ± 0.000 | 0.034 ± 0.000 | 0.109 ± 0.000 | 0.829 ± 0.000 |
| | | CIFAR100-LT | 0.2 | 0.138 ± 0.000 | 0.231 ± 0.000 | 0.660 ± 0.000 | 0.106 ± 0.000 | 0.203 ± 0.000 | 0.660 ± 0.000 | 0.105 ± 0.012 | 0.198 ± 0.010 | 0.660 ± 0.000 | 0.031 ± 0.000 | 0.112 ± 0.000 | 0.594 ± 0.000 |
| | | | 0.4 | 0.235 ± 0.000 | 0.291 ± 0.000 | 0.523 ± 0.000 | 0.056 ± 0.000 | 0.137 ± 0.000 | 0.523 ± 0.000 | 0.058 ± 0.006 | 0.145 ± 0.007 | 0.523 ± 0.000 | 0.040 ± 0.000 | 0.109 ± 0.000 | 0.556 ± 0.000 |
| | | | 0.6 | 0.295 ± 0.000 | 0.320 ± 0.000 | 0.431 ± 0.000 | 0.063 ± 0.000 | 0.166 ± 0.000 | 0.431 ± 0.000 | 0.067 ± 0.009 | 0.160 ± 0.010 | 0.431 ± 0.000 | 0.043 ± 0.000 | 0.142 ± 0.000 | 0.508 ± 0.000 |
| | | | 0.8 | 0.381 ± 0.000 | 0.360 ± 0.000 | 0.344 ± 0.000 | 0.121 ± 0.000 | 0.221 ± 0.000 | 0.344 ± 0.000 | 0.100 ± 0.016 | 0.193 ± 0.013 | 0.344 ± 0.000 | 0.080 ± 0.000 | 0.179 ± 0.000 | 0.432 ± 0.000 |
| | | | 1.0 | 0.403 ± 0.000 | 0.382 ± 0.000 | 0.269 ± 0.000 | 0.116 ± 0.000 | 0.234 ± 0.000 | 0.269 ± 0.000 | 0.129 ± 0.015 | 0.233 ± 0.010 | 0.269 ± 0.000 | 0.051 ± 0.000 | 0.165 ± 0.000 | 0.423 ± 0.000 |
| | ResNext50 | ImageNet-LT | 0.2 | 0.019 ± 0.000 | 0.061 ± 0.000 | 0.716 ± 0.000 | 0.073 ± 0.000 | 0.097 ± 0.000 | 0.716 ± 0.000 | 0.069 ± 0.006 | 0.095 ± 0.003 | 0.716 ± 0.000 | 0.013 ± 0.000 | 0.062 ± 0.000 | 0.627 ± 0.000 |
| | | | 0.4 | 0.032 ± 0.000 | 0.075 ± 0.000 | 0.592 ± 0.000 | 0.059 ± 0.000 | 0.089 ± 0.000 | 0.592 ± 0.000 | 0.056 ± 0.006 | 0.087 ± 0.003 | 0.592 ± 0.000 | 0.022 ± 0.000 | 0.068 ± 0.000 | 0.576 ± 0.000 |
| | | | 0.6 | 0.081 ± 0.000 | 0.103 ± 0.000 | 0.481 ± 0.000 | 0.045 ± 0.000 | 0.079 ± 0.000 | 0.481 ± 0.000 | 0.046 ± 0.002 | 0.081 ± 0.001 | 0.481 ± 0.000 | 0.020 ± 0.000 | 0.063 ± 0.000 | 0.525 ± 0.000 |
| | | | 0.8 | 0.171 ± 0.000 | 0.144 ± 0.000 | 0.309 ± 0.000 | 0.079 ± 0.000 | 0.115 ± 0.000 | 0.309 ± 0.000 | 0.086 ± 0.007 | 0.116 ± 0.002 | 0.309 ± 0.000 | 0.032 ± 0.000 | 0.080 ± 0.000 | 0.446 ± 0.000 |
| | | | 1.0 | 0.321 ± 0.000 | 0.198 ± 0.000 | 0.113 ± 0.000 | 0.230 ± 0.000 | 0.174 ± 0.000 | 0.113 ± 0.000 | 0.235 ± 0.008 | 0.175 ± 0.002 | 0.113 ± 0.000 | 0.082 ± 0.000 | 0.113 ± 0.000 | 0.318 ± 0.000 |
| | ResNext50(+P.B.) | ImageNet-LT | 0.2 | 0.017 ± 0.000 | 0.064 ± 0.000 | 0.653 ± 0.000 | 0.081 ± 0.000 | 0.100 ± 0.000 | 0.653 ± 0.000 | 0.084 ± 0.004 | 0.101 ± 0.002 | 0.653 ± 0.000 | 0.020 ± 0.000 | 0.070 ± 0.000 | 0.579 ± 0.000 |
| | | | 0.4 | 0.023 ± 0.000 | 0.068 ± 0.000 | 0.579 ± 0.000 | 0.062 ± 0.000 | 0.090 ± 0.000 | 0.579 ± 0.000 | 0.063 ± 0.004 | 0.089 ± 0.002 | 0.579 ± 0.000 | 0.013 ± 0.000 | 0.064 ± 0.000 | 0.535 ± 0.000 |
| | | | 0.6 | 0.065 ± 0.000 | 0.090 ± 0.000 | 0.508 ± 0.000 | 0.027 ± 0.000 | 0.074 ± 0.000 | 0.508 ± 0.000 | 0.027 ± 0.003 | 0.070 ± 0.002 | 0.508 ± 0.000 | 0.018 ± 0.000 | 0.067 ± 0.000 | 0.508 ± 0.000 |
| | | | 0.8 | 0.129 ± 0.000 | 0.121 ± 0.000 | 0.388 ± 0.000 | 0.042 ± 0.000 | 0.082 ± 0.000 | 0.388 ± 0.000 | 0.043 ± 0.003 | 0.083 ± 0.002 | 0.388 ± 0.000 | 0.039 ± 0.000 | 0.076 ± 0.000 | 0.444 ± 0.000 |
| | | | 1.0 | 0.236 ± 0.000 | 0.164 ± 0.000 | 0.224 ± 0.000 | 0.147 ± 0.000 | 0.134 ± 0.000 | 0.224 ± 0.000 | 0.147 ± 0.004 | 0.133 ± 0.002 | 0.224 ± 0.000 | 0.078 ± 0.000 | 0.100 ± 0.000 | 0.359 ± 0.000 |
| KNN | ResNet152 | Places365-LT | 0.2 | 0.159 ± 0.000 | 0.144 ± 0.000 | 0.492 ± 0.000 | 0.064 ± 0.000 | 0.099 ± 0.000 | 0.492 ± 0.000 | 0.047 ± 0.004 | 0.089 ± 0.002 | 0.492 ± 0.000 | 0.091 ± 0.000 | 0.116 ± 0.000 | 0.387 ± 0.000 |
| | | | 0.4 | 0.208 ± 0.000 | 0.166 ± 0.000 | 0.387 ± 0.000 | 0.061 ± 0.000 | 0.095 ± 0.000 | 0.387 ± 0.000 | 0.054 ± 0.001 | 0.097 ± 0.001 | 0.387 ± 0.000 | 0.048 ± 0.000 | 0.090 ± 0.000 | 0.448 ± 0.000 |
| | | | 0.6 | 0.270 ± 0.000 | 0.186 ± 0.000 | 0.317 ± 0.000 | 0.073 ± 0.000 | 0.111 ± 0.000 | 0.317 ± 0.000 | 0.068 ± 0.004 | 0.107 ± 0.002 | 0.317 ± 0.000 | 0.057 ± 0.000 | 0.098 ± 0.000 | 0.416 ± 0.000 |
| | | | 0.8 | 0.334 ± 0.000 | 0.214 ± 0.000 | 0.218 ± 0.000 | 0.120 ± 0.000 | 0.150 ± 0.000 | 0.218 ± 0.000 | 0.117 ± 0.004 | 0.149 ± 0.002 | 0.218 ± 0.000 | 0.065 ± 0.000 | 0.106 ± 0.000 | 0.367 ± 0.000 |
| | | | 1.0 | 0.437 ± 0.000 | 0.243 ± 0.000 | 0.081 ± 0.000 | 0.211 ± 0.000 | 0.185 ± 0.000 | 0.081 ± 0.000 | 0.202 ± 0.004 | 0.184 ± 0.001 | 0.081 ± 0.000 | 0.111 ± 0.000 | 0.152 ± 0.000 | 0.207 ± 0.000 |
| | ResNet18 | CIFAR10-LT | 0.2 | 0.017 ± 0.000 | 0.077 ± 0.000 | 0.927 ± 0.000 | 0.060 ± 0.000 | 0.164 ± 0.000 | 0.927 ± 0.000 | 0.098 ± 0.009 | 0.204 ± 0.008 | 0.927 ± 0.000 | 0.025 ± 0.000 | 0.085 ± 0.000 | 0.873 ± 0.000 |
| | | | 0.4 | 0.054 ± 0.000 | 0.145 ± 0.000 | 0.825 ± 0.000 | 0.082 ± 0.000 | 0.182 ± 0.000 | 0.825 ± 0.000 | 0.127 ± 0.008 | 0.220 ± 0.005 | 0.825 ± 0.000 | 0.024 ± 0.000 | 0.106 ± 0.000 | 0.742 ± 0.000 |
| | | | 0.6 | 0.152 ± 0.000 | 0.236 ± 0.000 | 0.672 ± 0.000 | 0.049 ± 0.000 | 0.133 ± 0.000 | 0.672 ± 0.000 | 0.054 ± 0.005 | 0.157 ± 0.005 | 0.672 ± 0.000 | 0.039 ± 0.000 | 0.124 ± 0.000 | 0.768 ± 0.000 |
| | | | 0.8 | 0.098 ± 0.000 | 0.188 ± 0.000 | 0.779 ± 0.000 | 0.022 ± 0.000 | 0.107 ± 0.000 | 0.779 ± 0.000 | 0.028 ± 0.002 | 0.106 ± 0.005 | 0.779 ± 0.000 | 0.029 ± 0.000 | 0.102 ± 0.000 | 0.810 ± 0.000 |
| | | | 1.0 | 0.244 ± 0.000 | 0.286 ± 0.000 | 0.584 ± 0.000 | 0.128 ± 0.000 | 0.206 ± 0.000 | 0.584 ± 0.000 | 0.153 ± 0.009 | 0.248 ± 0.007 | 0.584 ± 0.000 | 0.037 ± 0.000 | 0.123 ± 0.000 | 0.832 ± 0.000 |
| | | CIFAR100-LT | 0.2 | 0.138 ± 0.000 | 0.231 ± 0.000 | 0.660 ± 0.000 | 0.106 ± 0.000 | 0.203 ± 0.000 | 0.660 ± 0.000 | 0.088 ± 0.008 | 0.184 ± 0.008 | 0.660 ± 0.000 | 0.029 ± 0.000 | 0.108 ± 0.000 | 0.595 ± 0.000 |
| | | | 0.4 | 0.235 ± 0.000 | 0.291 ± 0.000 | 0.523 ± 0.000 | 0.056 ± 0.000 | 0.137 ± 0.000 | 0.523 ± 0.000 | 0.049 ± 0.003 | 0.135 ± 0.004 | 0.523 ± 0.000 | 0.049 ± 0.000 | 0.123 ± 0.000 | 0.554 ± 0.000 |
| | | | 0.6 | 0.295 ± 0.000 | 0.320 ± 0.000 | 0.431 ± 0.000 | 0.063 ± 0.000 | 0.166 ± 0.000 | 0.431 ± 0.000 | 0.073 ± 0.008 | 0.171 ± 0.008 | 0.431 ± 0.000 | 0.030 ± 0.000 | 0.113 ± 0.000 | 0.509 ± 0.000 |
| | | | 0.8 | 0.381 ± 0.000 | 0.360 ± 0.000 | 0.344 ± 0.000 | 0.121 ± 0.000 | 0.221 ± 0.000 | 0.344 ± 0.000 | 0.119 ± 0.012 | 0.210 ± 0.009 | 0.344 ± 0.000 | 0.078 ± 0.000 | 0.175 ± 0.000 | 0.435 ± 0.000 |
| | | | 1.0 | 0.403 ± 0.000 | 0.382 ± 0.000 | 0.269 ± 0.000 | 0.116 ± 0.000 | 0.234 ± 0.000 | 0.269 ± 0.000 | 0.145 ± 0.011 | 0.244 ± 0.008 | 0.269 ± 0.000 | 0.059 ± 0.000 | 0.163 ± 0.000 | 0.424 ± 0.000 |
| | ResNext50 | ImageNet-LT | 0.2 | 0.019 ± 0.000 | 0.061 ± 0.000 | 0.716 ± 0.000 | 0.073 ± 0.000 | 0.097 ± 0.000 | 0.716 ± 0.000 | 0.067 ± 0.004 | 0.094 ± 0.002 | 0.716 ± 0.000 | 0.011 ± 0.000 | 0.065 ± 0.000 | 0.626 ± 0.000 |
| | | | 0.4 | 0.032 ± 0.000 | 0.075 ± 0.000 | 0.592 ± 0.000 | 0.059 ± 0.000 | 0.089 ± 0.000 | 0.592 ± 0.000 | 0.052 ± 0.004 | 0.086 ± 0.002 | 0.592 ± 0.000 | 0.023 ± 0.000 | 0.068 ± 0.000 | 0.575 ± 0.000 |
| | | | 0.6 | 0.081 ± 0.000 | 0.103 ± 0.000 | 0.481 ± 0.000 | 0.045 ± 0.000 | 0.079 ± 0.000 | 0.481 ± 0.000 | 0.042 ± 0.001 | 0.080 ± 0.001 | 0.481 ± 0.000 | 0.023 ± 0.000 | 0.066 ± 0.000 | 0.524 ± 0.000 |
| | | | 0.8 | 0.171 ± 0.000 | 0.144 ± 0.000 | 0.309 ± 0.000 | 0.079 ± 0.000 | 0.115 ± 0.000 | 0.309 ± 0.000 | 0.088 ± 0.005 | 0.116 ± 0.002 | 0.309 ± 0.000 | 0.037 ± 0.000 | 0.079 ± 0.000 | 0.447 ± 0.000 |
| | | | 1.0 | 0.321 ± 0.000 | 0.198 ± 0.000 | 0.113 ± 0.000 | 0.230 ± 0.000 | 0.174 ± 0.000 | 0.113 ± 0.000 | 0.238 ± 0.005 | 0.176 ± 0.001 | 0.113 ± 0.000 | 0.086 ± 0.000 | 0.116 ± 0.000 | 0.318 ± 0.000 |
| | ResNext50(+P.B.) | ImageNet-LT | 0.2 | 0.017 ± 0.000 | 0.064 ± 0.000 | 0.653 ± 0.000 | 0.081 ± 0.000 | 0.100 ± 0.000 | 0.653 ± 0.000 | 0.083 ± 0.005 | 0.100 ± 0.003 | 0.653 ± 0.000 | 0.018 ± 0.000 | 0.069 ± 0.000 | 0.578 ± 0.000 |
| | | | 0.4 | 0.023 ± 0.000 | 0.068 ± 0.000 | 0.579 ± 0.000 | 0.062 ± 0.000 | 0.090 ± 0.000 | 0.579 ± 0.000 | 0.062 ± 0.005 | 0.089 ± 0.003 | 0.579 ± 0.000 | 0.015 ± 0.000 | 0.062 ± 0.000 | 0.535 ± 0.000 |
| | | | 0.6 | 0.065 ± 0.000 | 0.090 ± 0.000 | 0.508 ± 0.000 | 0.027 ± 0.000 | 0.074 ± 0.000 | 0.508 ± 0.000 | 0.028 ± 0.003 | 0.073 ± 0.001 | 0.508 ± 0.000 | 0.024 ± 0.000 | 0.069 ± 0.000 | 0.507 ± 0.000 |
| | | | 0.8 | 0.129 ± 0.000 | 0.121 ± 0.000 | 0.388 ± 0.000 | 0.042 ± 0.000 | 0.082 ± 0.000 | 0.388 ± 0.000 | 0.043 ± 0.005 | 0.083 ± 0.002 | 0.388 ± 0.000 | 0.038 ± 0.000 | 0.080 ± 0.000 | 0.444 ± 0.000 |
| | | | 1.0 | 0.236 ± 0.000 | 0.164 ± 0.000 | 0.224 ± 0.000 | 0.147 ± 0.000 | 0.134 ± 0.000 | 0.224 ± 0.000 | 0.148 ± 0.005 | 0.134 ± 0.002 | 0.224 ± 0.000 | 0.076 ± 0.000 | 0.101 ± 0.000 | 0.358 ± 0.000 |

Table 3: **Recalibration Results for LLM Classification.** For each dataset and atypicality quantile, the best results are marked in bold. We provide the standard errors next to the means over 10 random seeds.

| | | | | Uncalibrated ECE | RMSE | Accuracy | Content-Free ECE | RMSE | Accuracy | Atypicality-Aware ECE | RMSE | Accuracy |
|---|---|---|---|---|---|---|---|---|---|---|---|---|
| LLM | Alpaca7B | AG News | 0.25 | 0.180 ± 0.000 | 0.219 ± 0.000 | 0.681 ± 0.000 | 0.452 ± 0.000 | 0.411 ± 0.000 | 0.527 ± 0.000 | 0.070 ± 0.000 | 0.142 ± 0.000 | 0.760 ± 0.000 |
| | | | 0.50 | 0.165 ± 0.000 | 0.202 ± 0.000 | 0.671 ± 0.000 | 0.413 ± 0.000 | 0.370 ± 0.000 | 0.571 ± 0.000 | 0.027 ± 0.000 | 0.101 ± 0.000 | 0.775 ± 0.000 |
| | | | 0.75 | 0.169 ± 0.000 | 0.204 ± 0.000 | 0.657 ± 0.000 | 0.429 ± 0.000 | 0.364 ± 0.000 | 0.549 ± 0.000 | 0.030 ± 0.000 | 0.099 ± 0.000 | 0.752 ± 0.000 |
| | | | 1.00 | 0.202 ± 0.000 | 0.222 ± 0.000 | 0.580 ± 0.000 | 0.448 ± 0.000 | 0.350 ± 0.000 | 0.524 ± 0.000 | 0.028 ± 0.000 | 0.110 ± 0.000 | 0.715 ± 0.000 |
| | | IMDB | 0.25 | 0.023 ± 0.001 | 0.087 ± 0.002 | 0.887 ± 0.001 | 0.141 ± 0.000 | 0.194 ± 0.001 | 0.883 ± 0.001 | 0.011 ± 0.001 | 0.068 ± 0.002 | 0.927 ± 0.001 |
| | | | 0.50 | 0.024 ± 0.001 | 0.095 ± 0.002 | 0.851 ± 0.001 | 0.117 ± 0.002 | 0.185 ± 0.002 | 0.838 ± 0.002 | 0.014 ± 0.001 | 0.078 ± 0.002 | 0.920 ± 0.001 |
| | | | 0.75 | 0.051 ± 0.002 | 0.124 ± 0.002 | 0.795 ± 0.002 | 0.110 ± 0.001 | 0.181 ± 0.002 | 0.823 ± 0.002 | 0.009 ± 0.001 | 0.069 ± 0.001 | 0.895 ± 0.001 |
| | | | 1.00 | 0.063 ± 0.001 | 0.136 ± 0.001 | 0.756 ± 0.001 | 0.122 ± 0.001 | 0.201 ± 0.001 | 0.752 ± 0.001 | 0.021 ± 0.001 | 0.088 ± 0.002 | 0.883 ± 0.001 |
| | | TREC | 0.25 | 0.139 ± 0.000 | 0.355 ± 0.000 | 0.744 ± 0.000 | 0.286 ± 0.000 | 0.265 ± 0.000 | 0.776 ± 0.000 | 0.109 ± 0.000 | 0.266 ± 0.000 | 0.896 ± 0.000 |
| | | | 0.50 | 0.217 ± 0.000 | 0.444 ± 0.000 | 0.680 ± 0.000 | 0.314 ± 0.000 | 0.503 ± 0.000 | 0.672 ± 0.000 | 0.068 ± 0.000 | 0.148 ± 0.000 | 0.816 ± 0.000 |
| | | | 0.75 | 0.238 ± 0.000 | 0.482 ± 0.000 | 0.592 ± 0.000 | 0.464 ± 0.000 | 0.622 ± 0.000 | 0.520 ± 0.000 | 0.067 ± 0.000 | 0.198 ± 0.000 | 0.784 ± 0.000 |
| | | | 1.00 | 0.207 ± 0.000 | 0.451 ± 0.000 | 0.544 ± 0.000 | 0.528 ± 0.000 | 0.685 ± 0.000 | 0.432 ± 0.000 | 0.150 ± 0.000 | 0.199 ± 0.000 | 0.696 ± 0.000 |

# D  Recalibration

Through all our recalibration results, we first split the test set into two equally sized calibration and test splits. Then, we fit the recalibration method using the calibration split and compute the performance on the test split. We run all our experiments with 10 different random seeds.

## D.1  Temperature Scaling

To perform temperature scaling [GPSW17], we use the calibration set to fit the temperature parameter. To perform the optimization, we use the LBFGS [LN89] algorithm from PyTorch with strong Wolfe line search, following [GPSW17]. Namely, we optimize the parameter $\tau$ with

$$\hat{\mathbb{P}}_{\text{TS}}(X) = \text{Softmax}(f(\boldsymbol{X})/\tau) \tag{9}$$

and then use it during inference to rescale the logits produced by $f$. We use $0.1$ learning rate and $3000$ maximum iterations across all experiments and initialize the temperature value as $1$, although find that TS is pretty robust to the choice of hyperparameters.

## D.2  Atypicality-Aware Recalibration

Here we describe the implementation details For Atypicality-Aware Recalibration (AAR). We formulate AAR with:

$$\log \hat{\mathbb{P}}_{\text{AAR}}(Y|X) \propto \phi(a(X)) \log \hat{\mathbb{P}}(Y|X) + S_Y, \tag{10}$$

In total, this gives us $|\mathcal{Y}| + 3$ parameters. Using exactly the same setting as TS, we use LBFGS with a strong Wolfe search to optimize the three parameters, with the same splits as temperature scaling. We normalize the atypicality values (subtract the mean and divide by the standard deviation of the calibration set) for numerical stability. We use the same hyperparameters as TS (with $0.1$ learning rate and $3000$ maximum iterations) without any modification across all experiments, initialize $c_0, c_1, c_2$ as $0$ and $S_Y$ parameters as $1$. We run the recalibration procedure on a CPU with precomputed logits.

### D.2.1  Adaptive Temperature Scaling

For AdaTS [JPL$^+$23] we use the implementation provided with the paper [7]. We identically use the hyperparameters and the architecture provided in the paper and their repository. They use an encoder and decoder architecture with $[1024, 512, 512]$ hidden units each, and a temperature predictor network with $[128, 128]$ hidden units. They use an Adam Optimizer with a learning rate of $5e - 4$ with $128$ batch size.

## D.3  Conformal Prediction

We follow the presentation in [AB21, ABMJ20]. Let $\pi(X)$ be the permutation of $\mathcal{Y} = \{1, \dots, C\}$ that sorts $\hat{\mathbb{P}}(Y = c|X)$, i.e. the predicted probabilities for each class $c$. We define a score function

$$s(x, y) = \sum_{j=1}^{c} \hat{\mathbb{P}}(Y = j|X), \text{ where } y = \pi_c. \tag{11}$$

This means greedily including classes until the set contains the true label, and using the cumulative sum of the probabilities as the score function. We compute all of the scores for the calibration set, $S_{\text{calib}} = \{s(x_1, y_1), ..., s(x_N, y_N)\}$, we the $\frac{\lceil (N+1)(1-\alpha) \rceil}{N}$ th quantile of the scores, $\hat{q}$. Then, the prediction set is defined as

$$\mathcal{C}(x) = \{y : s(x, y) \leq \hat{q}\} \tag{12}$$

We can further add randomization to the procedure where we have the prediction set function to be $\mathcal{C}(x, u) : \mathcal{X} \times [0, 1]$ for randomization purposes to satisfy exact coverage. We refer to [VGS05, ABMJ20, AB21] for a more thorough presentation.

---

[7]https://github.com/thwjoy/adats

RAPS is a variant of APS that regularizes the set sizes. They modify the scoring function to add a regularization term. This is controlled by the test size offset $k_{reg}$ that controls the value beyond which the regularization is applied, and the $\lambda_{reg}$ gives the strength of the regularization. To fit the $k_{reg}, \lambda_{reg}$ parameters in RAPS, we follow the procedure in [ABMJ20] to fit both parameters. Namely, we fit $k_{reg}$ by Algorithm 4 in their paper that leverages the set sizes in the calibration set, and we fit $\lambda_{reg}$ by the largest regularization parameter that achieves the smallest set sizes, searched over a grid of $\{0.001, 0.01, 0.1, 0.2, 0.5\}$ following their presentation.

### D.4 Atypicality-Aware Conformal Prediction

We have a simple discrete grouping scheme to make conformal prediction atypicality aware. Namely, we group points using their atypicality and confidence percentiles and fit individual thresholds. Concretely, we construct a dataset of $\mathcal{D}_{AA} = \{(c_i, c_{i+1}], (a_j, a_{j+1}], \hat{q}_{i,j}\}_{i,j \in [N]}$ using the calibration set where $(c_i, c_{i+1}]$ denotes the confidence range for quantile $i$, $(a_j, a_{j+1}]$ denotes the atypicality range for quantile $j$ and $\hat{q}_{i,j}$ denotes the threshold fitted to the group specified by these intervals. We let $N = 6$ as the number of groups, and in total, we end up with 36 thresholds. At test time, we check the quantile of the confidence and atypicality of a point and use the corresponding temperature. For AA-RAPS, we use the same $k_{reg}$ and $\lambda_{reg}$ values as was found with the RAPS procedure. For practical purposes, we do not allow zero sets (prediction sets at least include the top prediction).

We would like to make the remark that sometimes the marginal coverage can exceed the desired value (e.g. Figure 9). This is often because the underlying model is already very confident for a majority of data points (e.g. More than half of the data points have 92% confidence). The gains we provide are often for points with lower confidence regions, as the coverage is not satisfied in those regions.

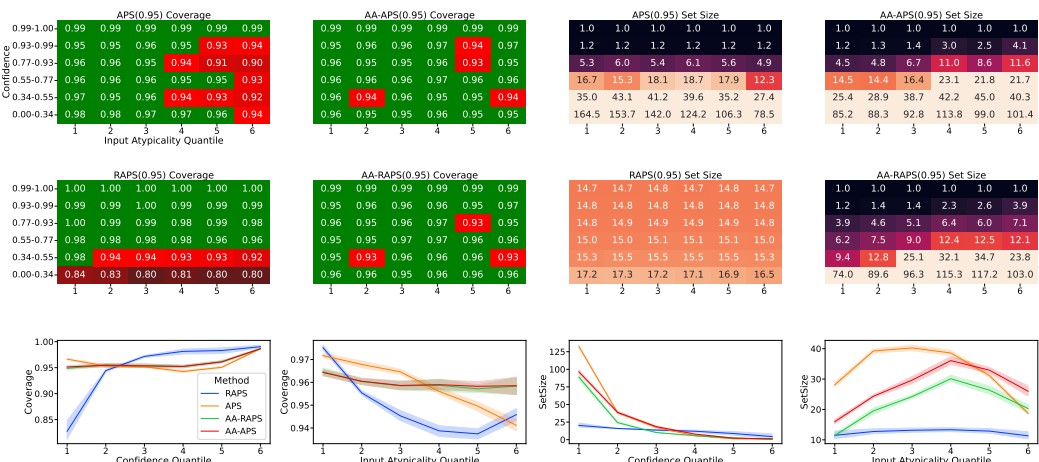

Figure 8: **Atypicality-Aware Conformal Prediction for ResNet18 and ImageNet.** Target coverage rate is 95%.

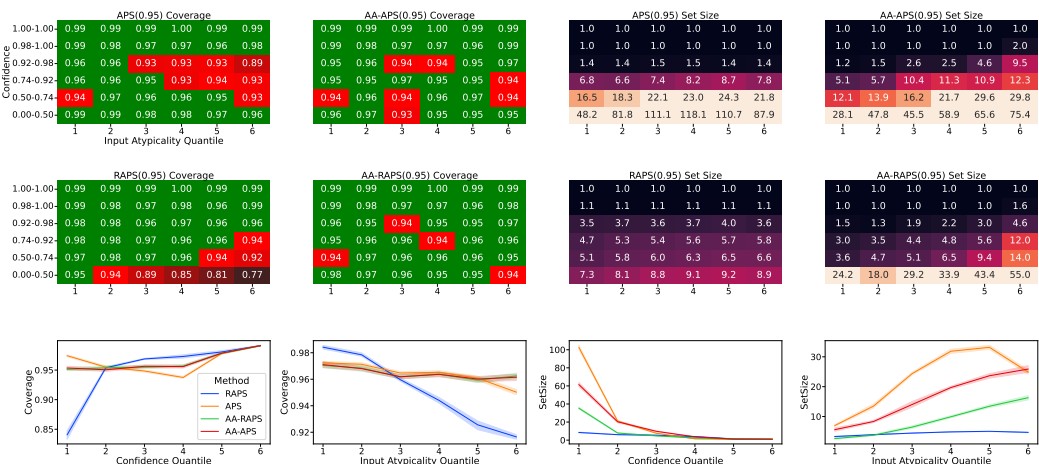

Figure 9: **Atypicality-Aware Conformal Prediction for ResNet152 and ImageNet.** Target coverage rate is 95%.

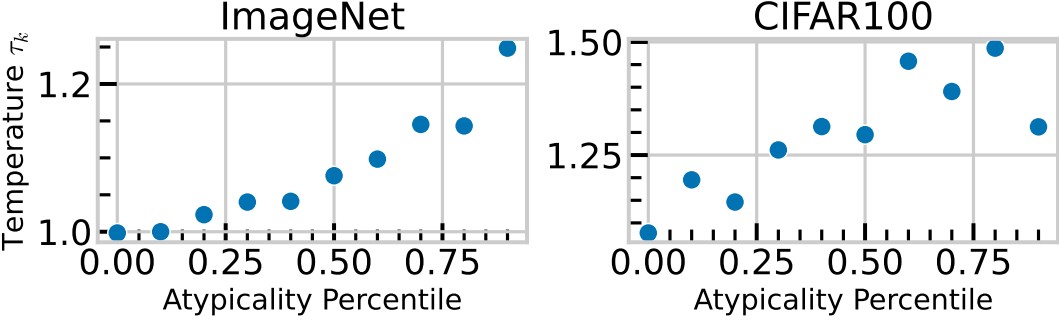

Figure 10: **Fitted Temperature vs Atypicality.** We observe a monotonically increasing relationship between the atypicality of a group and the temperature parameter fitted to that group with TS.

# E   Tables for Results

Here, we present the table version of the results in Figure 3. Tables 1,2 contain the ECE analysis.

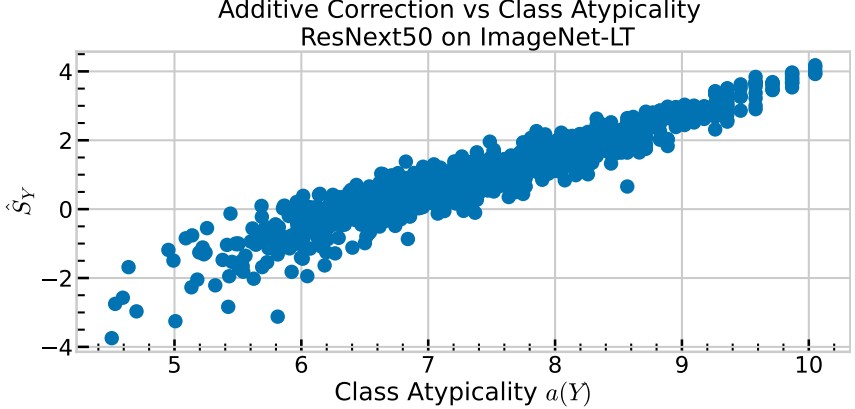

Figure 11: **Fitted Additive Correction Factor vs Class Atypicality.** We observe a monotonically increasing relationship between the atypicality of a class and the additive correction parameter fitted to that class with AAR.

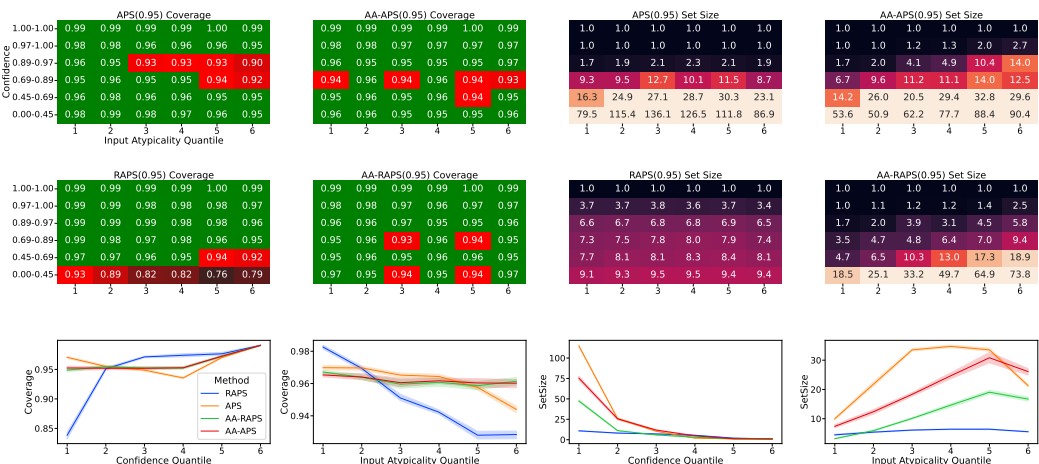

Figure 12: **Atypicality-Aware Conformal Prediction for ResNet50 and ImageNet.** Target coverage rate is 95%.

## F   Fitzpatrick17k and Skin Lesion Classification

We use the training script from [GHS⁺21] to finetune models on the Fitzpatrick17k dataset. We train the models for 50 epochs, fixing the backbone and training only the probe on top of the penultimate layer. The probe consists of 2 layers, one layer of 256 units followed by ReLU and Dropout with probability 0.4, followed by the classifier layer with an output dimensionality of 9. We use an Adam optimizer with a 0.0001 learning rate.

The entire dataset consists of $16,577$ images, where the potential labels are: $10,886$ inflammatory, $1,352$ malignant epidermal, $1,194$ genodermatoses, $1,067$ benign dermal, $931$ benign epidermal, $573$ malignant melanoma, $236$ benign melanocyte, $182$ malignant cutaneous lymphoma, and $156$ malignant dermal. We split the dataset into 3 sets (Training $(0.5)$, Validation $(0.25)$, and Test $(0.25)$). We use the validation set as the calibration set and perform the experiments with 10 random splits.

# G Proofs

## G.1 Detailed derivation of the claim on Page 5

When $\mathbb{P}_1(X) \leq 1/2$, the signed calibration error at level $u \in (1/2, 1)$ becomes $u - \mathbb{P}(Y = -1 \mid \hat{\mathbb{P}}_{-1}(X) = u) = u - \mathbb{P}(Y = -1 \mid \hat{\mathbb{P}}_1(-X) = u) = u - \mathbb{P}(Y = 1 \mid \hat{\mathbb{P}}_1(X) = u)$.

The last inequality is due to symmetry. More specifically, we claim $(X, Y) \stackrel{d}{=} (-X, -Y)$, where the notation $\stackrel{d}{=}$ denotes equal in distribution. In fact, as $X \stackrel{d}{=} -X$, it suffices to show that for any $y \in \{-1, 1\}$, and $x \in \mathbb{R}^d$, we have

$$\mathbb{P}(Y = y \mid X = x) = \mathbb{P}(-Y = y \mid -X = x).$$

When $y = -1$, the right hand side

$$\mathbb{P}(-Y = -1 \mid -X = x) = \mathbb{P}(Y = 1 \mid X = -x) = \sigma(\langle \beta^*, -x \rangle)$$
$$= 1 - \sigma(\langle \beta^*, x \rangle) = 1 - \mathbb{P}(Y = 1 \mid X = x) = \mathbb{P}(Y = -1 \mid X = x).$$

Similarly, when $y = 1$,

$$\mathbb{P}(-Y = 1 \mid -X = x) = \mathbb{P}(Y = -1 \mid X = -x) = 1 - \sigma(\langle \beta^*, -x \rangle)$$
$$= \sigma(\langle \beta^*, x \rangle) = \mathbb{P}(Y = 1 \mid X = x).$$

We complete the proof.

## G.2 Proof of Theorem 3.1

**Theorem G.1** (Restatement of Theorem 3.1). *Consider the data generative model with the algorithm described in Section 3.2. For any $K > 1$, suppose we consider the quantiles of $a(X)$, $a_1, a_2, ..., a_K, a_{K+1}$ such that $\mathbb{P}(a(X) \in (a_k, a_{k+1}]) = 1/K$ for $k \in [K]$. In addition, we assume $\|\beta^*\| \leq c_0$, and $d/n = \kappa$ for some sufficiently small $c_0, \kappa > 0$. Then for sufficiently large $n$, we have*

$$\mathbb{E}_u[u - \mathbb{P}(Y = 1 \mid \hat{\mathbb{P}}_1(X) = u) \mid a(X) \in [a_{k-1}, a_k]] >$$
$$\mathbb{E}_u[u - \mathbb{P}(Y = 1 \mid \hat{\mathbb{P}}_1(X) = u) \mid a(X) \in (a_k, a_{k+1}]],$$

*for $k = 2, .., K$.*

*Proof.* Following [BMWX21a], we have

$$u - \mathbb{P}(Y = 1 \mid \hat{\mathbb{P}}_1(X) = u) = u - \mathbb{E}_Z[\sigma(\frac{\|\beta^*\|}{\|\hat{\beta}\|} \cos \hat{\theta} \cdot \sigma^{-1}(u)) + \sin \hat{\theta} \cdot \|\beta^*\| Z],$$

where $\cos \hat{\theta} = \frac{\hat{\beta}^\top \beta^*}{\|\hat{\beta}\| \cdot \|\beta^*\|}$ and $Z \sim N(0, 1)$.

According to the results in Section 2.2 of [SC19], we have $\|\hat{\beta}\| \to R^* = R^*(\kappa, \beta^*)$ and $\cos \hat{\theta} \to c^* = c^*(\kappa, \beta^*)$, for two quantities $R^*$ and $c^*$ that depend on $\kappa$ and $\beta^*$. We then have

$$u - \mathbb{P}(Y = 1 \mid \hat{\mathbb{P}}_1(X) = u) \to u - \mathbb{E}_Z[\sigma(\frac{\|\beta^*\|}{R^*} c^* \cdot \sigma^{-1}(u)) + \sqrt{1 - c^{*2}} \cdot \|\beta^*\| Z].$$

Using the proof of Theorem 3 in [BMWX21a], we have that

$$u - \mathbb{P}(Y = 1 \mid \hat{\mathbb{P}}_1(X) = u) = C_\kappa(u) \cdot \kappa + o(\kappa),$$

where

$$C_\kappa(u) = c_1 \sigma'(\sigma^{-1}(u)) \cdot \sigma^{-1}(u) - c_2 \sigma''(\sigma^{-1}(u)),$$

for two positive constants $c_1, c_2$.

As a result, we have

$$u - \mathbb{P}(Y = 1 \mid \hat{\mathbb{P}}_1(X) = u) \geq 0 \tag{13}$$

In addition, since when $z \in [-1, 1]$, $z \cdot \sigma'(z)$ and $-\sigma''(z)$ are both increasing, we then have $C_\kappa(u)$ increasing for $\hat{\beta}^\top x = \sigma^{-1}(u) \in (-1, 1)$.

**Proving the result for $\{k = 2, \ldots, K-1\}$** In addition, by our model assumption $x \sim N(0, I_d)$, we have that $\|x\|$ and $\frac{x}{\|x\|}$ are independent, and $\frac{x}{\|x\|} \sim S$ where $S$ is a uniform distribution on the sphere in the $d$-dimensional space. As the monotonic transformations will not change the events defined by quantiles, and $\exp(-\|x\|^2/2)$ is a monotonic function in $\|x\|$, for the simplicity of presentation we use $a(X) = \|X\|$ in the rest of this proof. As a result, given $\|x\| = a$, we have

$$\hat{\beta}^\top x \mid \|x\| = a \overset{d}{=} a \cdot \hat{\beta}^\top S = a \cdot \|\hat{\beta}\| \cdot S_1,$$

where $S_1$ is the first coordinate of $S$.

Consequently, if we further condition on the event where $\hat{\beta}^\top x > 0$ (as we assume $u > 0$ throughout Section 3.2), we have

$$\hat{\beta}^\top x \overset{d}{=} a \cdot \|\hat{\beta}\| \cdot S_1 \mid S_1 > 0 \overset{d}{=} a \cdot \|\hat{\beta}\| \cdot \frac{Z_1}{\sqrt{Z_1^2 + Q}} \to a \cdot R^* \cdot \frac{Z_1}{\sqrt{Z_1^2 + Q}},$$

where $Q \sim \chi_{p-1}^2$, $Z_1 \sim N(0,1)$ and they are independent.

Due to the monotonicity of $C_\kappa(u)$ on $u$, we have that for any $a_1 > a_2$,

$$C_\kappa(u) \mid \|x\| = a_1 \overset{d}{>} C_\kappa(u) \mid \|x\| = a_2,$$

where the notation $\overset{d}{>}$ denotes stochastic dominance.

Consequently, we have

$$\mathbb{E}_u[u - \mathbb{P}(Y = 1 \mid \hat{\mathbb{P}}_1(X) = u) \mid a(X) \in [a_{k-1}, a_k]] < \mathbb{E}_u[u - \mathbb{P}(Y = 1 \mid \hat{\mathbb{P}}_1(X) = u) \mid a(X) \in (a_k, a_{k+1}]],$$

for $k = 2, .., K - 1$.

**Proving the result for $k = K$** To complete the proof, it suffices to show that the inequality is also true for $K$th quantile:

$$\mathbb{E}_u[u - \mathbb{P}(Y = 1 \mid \hat{\mathbb{P}}_1(X) = u) \mid a(X) \in [a_{K-1}, a_K]] < \mathbb{E}_u[u - \mathbb{P}(Y = 1 \mid \hat{\mathbb{P}}_1(X) = u) \mid a(X) \in (a_k, a_{k+1}]],$$

which is equivalent to

$$\mathbb{E}_u[(u - \mathbb{P}(Y = 1 \mid \hat{\mathbb{P}}_1(X) = u)) \cdot \mathbf{1}\{a(X) \in [a_{K-1}, a_K]\}] < \mathbb{E}_u[(u - \mathbb{P}(Y = 1 \mid \hat{\mathbb{P}}_1(X) = u)) \cdot \mathbf{1}\{a(X) \in (a_k, a_{k+1}]\}].$$

In the above inequality, the right hand side can be decomposed into

$$\begin{aligned}
&\mathbb{E}_u[(u - \mathbb{P}(Y = 1 \mid \hat{\mathbb{P}}_1(X) = u)) \cdot \mathbf{1}\{a(X) \in (a_k, a_{k+1}]\}] \\
=&\mathbb{E}_u[(u - \mathbb{P}(Y = 1 \mid \hat{\mathbb{P}}_1(X) = u)) \cdot \mathbf{1}\{a(X) \in [a_K, 2p]\}] \\
&+ \mathbb{E}_u[(u - \mathbb{P}(Y = 1 \mid \hat{\mathbb{P}}_1(X) = u)) \cdot \mathbf{1}\{a(X) \in [2p, a_{K+1}]\}].
\end{aligned}$$

Denote the $\alpha$ quantile of $\chi_p^2$ by $\chi_{\alpha,p}^2$. We then have $a_k = \chi_{\frac{k}{K+1},p}^2$. We further decompose the equation into

$$\begin{aligned}
&\mathbb{E}_u[(u - \mathbb{P}(Y = 1 \mid \hat{\mathbb{P}}_1(X) = u)) \cdot \mathbf{1}\{a(X) \in [a_K, 2p]\}] \\
=&\mathbb{E}_u[(u - \mathbb{P}(Y = 1 \mid \hat{\mathbb{P}}_1(X) = u)) \cdot \mathbf{1}\{a(X) \in [a_K, \chi_{\frac{K+\delta}{K+1},p}^2]\}] \\
&+ \mathbb{E}_u[(u - \mathbb{P}(Y = 1 \mid \hat{\mathbb{P}}_1(X) = u)) \cdot \mathbf{1}\{a(X) \in [\chi_{\frac{K+\delta}{K+1},p}^2, 2p]\}].
\end{aligned}$$

In the following, we proceed to prove

$$\mathbb{E}_u[(u - \mathbb{P}(Y = 1 \mid \hat{\mathbb{P}}_1(X) = u)) \cdot \mathbf{1}\{a(X) \in [\chi_{\frac{K+\delta}{K+1},p}^2, 2p]\}] > \mathbb{E}_u[(u - \mathbb{P}(Y = 1 \mid \hat{\mathbb{P}}_1(X) = u)) \cdot \mathbf{1}\{a(X) \in [a_{K-1}, a_K]\}].$$
$$(14)$$

We now use the approximation of the chi-square quantile: when $p \to \infty$, we have

$$a_K = \frac{1}{2}(z_{\frac{K}{K+1}} + \sqrt{2p})^2 + o(1), \text{ and } \chi_{\frac{K+\delta}{K+1},p}^2 = \frac{1}{2}(z_{\frac{K+\delta}{K+1}} + \sqrt{2p})^2 + o(1),$$

where $z_\alpha$ denotes the $\alpha$-quantile of a standard normal random variable.

Then

$$\chi^2_{\frac{K+\delta}{K+1},p} - a_K = \frac{1}{2}(z_{\frac{K+\delta}{K+1}} - z_{\frac{K}{K+1}})(z_{\frac{K+\delta}{K+1}} + z_{\frac{K}{K+1}} + 2\sqrt{2p}).$$

Using the fact that $z_{1-\frac{1}{K}} = \sqrt{2\log K} + o(1)$ for $K \to \infty$, then we have

$$z_{\frac{K+\delta}{K+1}} - z_{\frac{K}{K+1}} = \frac{-\log(1-\delta)}{\sqrt{2\log K}} + o(1).$$

In addition, for any $a \in [\chi^2_{\frac{K+\delta}{K+1},p}, 2p]$ and $a' \in [a_{K-1}, a_K]$, we have

$\mathbb{E}_u[(u - \mathbb{P}(Y = 1 \mid \hat{\mathbb{P}}_1(X) = u)) \mid a(X) = a] - \mathbb{E}_u[(u - \mathbb{P}(Y = 1 \mid \hat{\mathbb{P}}_1(X) = u)) \mid a(X) = a']$
$\geq C(z_{\frac{K+\delta}{K+1}} - z_{\frac{K}{K+1}}),$

for some universal constant $C$.

Therefore

$\mathbb{E}_u[(u - \mathbb{P}(Y = 1 \mid \hat{\mathbb{P}}_1(X) = u)) \mid a(X) \in [\chi^2_{\frac{K+\delta}{K+1},p}, 2p]] - \mathbb{E}_u[(u - \mathbb{P}(Y = 1 \mid \hat{\mathbb{P}}_1(X) = u)) \mid a(X) \in [a_{K-1}, a_K]]$
$\geq C(z_{\frac{K+\delta}{K+1}} - z_{\frac{K}{K+1}}).$

Then

$\mathbb{E}_u[(u - \mathbb{P}(Y = 1 \mid \hat{\mathbb{P}}_1(X) = u)) \cdot \mathbf{1}\{a(X) \in [\chi^2_{\frac{K+\delta}{K+1},p}, 2p]\}]$
$= \mathbb{E}_u[(u - \mathbb{P}(Y = 1 \mid \hat{\mathbb{P}}_1(X) = u)) \mid a(X) \in [\chi^2_{\frac{K+\delta}{K+1},p}, 2p]] \cdot \mathbb{P}(a(X) \in [\chi^2_{\frac{K+\delta}{K+1},p}, 2p])$
$\geq \left( \mathbb{E}_u[(u - \mathbb{P}(Y = 1 \mid \hat{\mathbb{P}}_1(X) = u)) \mid a(X) \in [a_{K-1}, a_K]] + C(z_{\frac{K+\delta}{K+1}} - z_{\frac{K}{K+1}}) \right) \cdot (\frac{1}{K} - \frac{\delta}{K+1} + o(\frac{\delta}{K+1}))$
$= \mathbb{E}_u[(u - \mathbb{P}(Y = 1 \mid \hat{\mathbb{P}}_1(X) = u)) \cdot \mathbf{1}\{a(X) \in [a_{K-1}, a_K\}]$
$\quad + C(z_{\frac{K+\delta}{K+1}} - z_{\frac{K}{K+1}}) - (1 + o(1))\frac{\delta}{K+1} \cdot \mathbb{E}_u[(u - \mathbb{P}(Y = 1 \mid \hat{\mathbb{P}}_1(X) = u)) \mid a(X) \in [a_{K-1}, a_K]].$

The last equality uses the fact that $\mathbb{P}(a(X) \in [a_{K-1}, a_K]) = 1/K$, and therefore

$$\mathbb{E}_u[(u - \mathbb{P}(Y = 1 \mid \hat{\mathbb{P}}_1(X) = u)) \mid a(X) \in [a_{K-1}, a_K] \cdot \frac{1}{K} = \mathbb{E}_u[(u - \mathbb{P}(Y = 1 \mid \hat{\mathbb{P}}_1(X) = u)) \cdot \mathbf{1}\{a(X) \in [a_{K-1}, a_K\}]$$

Then use the fact that $|\mathbb{E}_u[(u - \mathbb{P}(Y = 1 \mid \hat{\mathbb{P}}_1(X) = u)) \mid a(X) \in [a_{K-1}, a_K]]| = O(1)$ and we choose $\delta = o(1/\log K)$ so

$$\frac{\delta}{K} = o(|\frac{\log(1-\delta)}{\sqrt{\log K}}|).$$

Consequently,

$$C(z_{\frac{K+\delta}{K+1}} - z_{\frac{K}{K+1}}) - (1+o(1))\frac{\delta}{K+1} \cdot \mathbb{E}_u[(u - \mathbb{P}(Y = 1 \mid \hat{\mathbb{P}}_1(X) = u)) \mid a(X) \in [a_{K-1}, a_K]] > 0,$$

which implies

$$\mathbb{E}_u[(u - \mathbb{P}(Y = 1 \mid \hat{\mathbb{P}}_1(X) = u)) \cdot \mathbf{1}\{a(X) \in [\chi^2_{\frac{K+\delta}{K+1},p}, 2p]\}] \geq \mathbb{E}_u[(u - \mathbb{P}(Y = 1 \mid \hat{\mathbb{P}}_1(X) = u)) \cdot \mathbf{1}\{a(X) \in [a_{K-1}, a_K\}].$$

Combining with equation 13, we prove equation 14 and complete the proof. $\qquad \square$

### G.3  Theoretical Justification of the calibration improvement using the atypicality score

In this section, we provide the theoretical justification to understand why incorporating the atypicality score will improve calibration. In particular, we consider the binary classification problem with prediction $f : X \in [0,1]$ indicating the predicted probability of $Y = 1$ given $X = x$.

For a predictor $f$, let us denote its conditional calibration error at an atypicality level $\gamma$ by $\mathrm{CE}_\gamma(f) = \mathbb{E}[(f(X) - \mathbb{E}[Y|f(X)])^2 | a(X) = \gamma]$.

**Theorem G.2.** *Consider the same setting as Theorem 3.1. Suppose the temperature function $\hat\tau(a(X)) = \arg\min_\tau \mathbb{E}[l(Y, \mathrm{Softmax}(f(X)/\tau(a(X))))]$ with $l$ being the cross entropy loss, and let $\hat{\mathbb{P}}_{\mathrm{AAR}}(X) = \mathrm{Softmax}(f(X)/\hat\tau(a(X)))$. Then*

$$\mathrm{CE}_\gamma(\hat{\mathbb{P}}_{\mathrm{AAR}}) \le \min\{\mathrm{CE}_\gamma(\hat{\mathbb{P}}_{\mathrm{TS}}), \mathrm{CE}_\gamma(f)\}. \tag{15}$$

Proof: For a prediction function $f$, we first define the conditional mean squared error of $f$ at an atypicality level $\gamma$ by $\mathrm{MSE}_\gamma(f) = \mathbb{E}[(f(X) - Y)^2 | a(X) = \gamma]$, then we have

$$\mathrm{MSE}_\gamma(f) - CE_\gamma(f) = \mathbb{E}[(f(X) - Y)^2 | a(X) = \gamma] - \mathbb{E}[(f(X) - \mathbb{E}[Y | f(X), a(X) = \gamma])^2 | a(X) = \gamma]$$
$$= \mathbb{E}[(\mathbb{E}[Y | f(X), a(X) = \gamma] - Y) \cdot (2f(X) - \mathbb{E}[Y | f(X), a(X) = \gamma] - Y) | a(X) = \gamma]$$
$$= \mathbb{E}[(\mathbb{E}[Y | f(X), a(X) = \gamma] - Y) \cdot (\mathbb{E}[Y | f(X), a(X) = \gamma] - Y) | a(X) = \gamma]$$
$$\quad + 2\mathbb{E}[(\mathbb{E}[Y | f(X), a(X) = \gamma] - Y) \cdot (f(X) - \mathbb{E}[Y | f(X), a(X) = \gamma])) | a(X) = \gamma]$$

Since

$$\mathbb{E}[Y\mathbb{E}[Y | f(X), a(X) = \gamma] | a(X) = \gamma]$$
$$= \mathbb{E}_{f(X)|a(X)=\gamma}\mathbb{E}[Y\mathbb{E}[Y | f(X), a(X) = \gamma] | f(X), a(X) = \gamma]]$$
$$= \mathbb{E}[(\mathbb{E}[Y | f(X), a(X) = \gamma])^2 | a(X) = \gamma],$$

we have

$$\mathbb{E}[(\mathbb{E}[Y | f(X), a(X) = \gamma] - Y) \cdot (f(X) - \mathbb{E}[Y | f(X), a(X) = \gamma])) | a(X) = \gamma] = 0,$$

and therefore

$$\mathrm{MSE}_\gamma(f) - CE_\gamma(f) = \mathbb{E}[(\mathbb{E}[Y | f(X), a(X) = \gamma] - Y)^2 | a(X) = \gamma]$$

Now that $\hat{\mathbb{P}}_{AAR}(f(x), a(x))$ is monotonic on the $\hat{\mathbb{P}}(x)$, we have

$$\mathbb{E}[Y | f(x), a(X) = \gamma] = \mathbb{E}[Y | \hat{\mathbb{P}}_{AAR}(f(x), a(X)), a(X) = \gamma],$$

implying

$$\mathrm{MSE}_\gamma(\hat{\mathbb{P}}_{AAR}) - CE_\gamma(\hat{\mathbb{P}}_{AAR}) = \mathrm{MSE}_\gamma(\hat{\mathbb{P}}) - CE_\gamma(\hat{\mathbb{P}}). \tag{16}$$

Similarly, we have

$$\mathrm{MSE}_\gamma(\hat{\mathbb{P}}_{TS}) - CE_\gamma(\hat{\mathbb{P}}_{TS}) = \mathrm{MSE}_\gamma(\hat{\mathbb{P}}) - CE_\gamma(\hat{\mathbb{P}}). \tag{17}$$

In the following, we will show that

$$\mathrm{MSE}_\gamma(\hat{\mathbb{P}}_{AAR}) < \min\{\mathrm{MSE}_\gamma(\hat{\mathbb{P}}_{TS}), \mathrm{MSE}_\gamma(\hat{\mathbb{P}})\}. \tag{18}$$

First, as we consider the binary classification setting, with $l$ being the cross-entropy loss, we have

$$l(Y, \mathrm{Softmax}(f(X)/\tau(a(X)))) = Y\log(\sigma(f(X)/\tau(a(X)))) + (1-Y)\log(1 - \sigma(f(X)/\tau(a(X)))),$$

where $\sigma(x) = 1/(1 + e^x)$.

Then, by the definition of $\hat\tau(a(X))$, we have that

$$\hat\tau(a(X)) = \arg\min_\tau \mathbb{E}[Y\log(\sigma(f(X)/\tau(a(X))) + (1 - Y)\log(1 - \sigma(f(X)/\tau(a(X))))]$$
$$= \arg\min_\tau \mathbb{E}[\mathbb{E}[Y\log(\sigma(f(X)/\tau(a(X))) + (1 - Y)\log(1 - \sigma(f(X)/\tau(a(X))) | a(X)]].$$

Taking the derivative on the last line and setting it to zero, we have

$$\mathbb{E}[\frac{Y}{\sigma(f(X)/\hat{\tau}(a(X)))} - \frac{1-Y}{1-\sigma(f(X)/\hat{\tau}(a(X)))} \mid a(X)] = 0,$$

implying

$$\mathbb{E}[\sigma(f(X)/\hat{\tau}(a(X))) \mid a(X)] = \mathbb{E}[Y \mid a(X)].$$

This makes the derivative of $\mathbb{E}[(Y - \sigma(f(X)/\tau(a(X))))^2 \mid a(X)]$ zero and therefore $\hat{\tau}(a(X))$ is also a minimizer of $\mathbb{E}[(Y - \sigma(f(X)/\tau(a(X))))^2 \mid a(X)]$:

$$\hat{\tau}(a(X)) = \arg\min_{\tau} \mathbb{E}[Y\log(\sigma(f(X)/\tau(a(X)))+(1-Y)\log(1-\sigma(f(X)/\tau(a(X))))] = \arg\min_{\tau} \mathbb{E}[(Y-\sigma(f(X)/\tau(a(X))))^2$$

Letting $g(\gamma) = \arg\min_c \mathbb{E}[(Y - \sigma(f(X)/c))^2 \mid a(X) = \gamma]$, we have that

$$g(a(X)) = \arg\min_{\tau} \mathbb{E}[(Y - \sigma(f(X)/\tau(a(X))))^2 \mid a(X)],$$

and therefore

$$g(a(X)) = \arg\min_{\tau} \mathbb{E}[\mathbb{E}[(Y - \sigma(f(X)/\tau(a(X))))^2 \mid a(X)] = \hat{\tau}(a(X)).$$

As a result,

$$
\begin{aligned}
\text{MSE}_{\gamma}(\hat{\mathbb{P}}_{AAR}) &= \mathbb{E}[(\hat{\mathbb{P}}_{AAR}(X) - Y)^2 \mid a(X) = \gamma] \\
&= \mathbb{E}[(\hat{\mathbb{P}}_{AAR}(X) - Y)^2 \mid a(X) = \gamma] \\
&= \mathbb{E}[(\sigma(f(X)/\hat{\tau}(a(X))) - Y)^2 \mid a(X) = \gamma] \\
&= \mathbb{E}[(\text{Softmax}(\hat{\mathbb{P}}(X)/g(a(X))) - Y)^2 \mid a(X) = \gamma] \\
&= \arg\min_c \mathbb{E}[(\sigma(f(X)/c - Y)^2 \mid a(X) = \gamma] \\
&\leq \mathbb{E}[(\sigma(f(X) - Y)^2 \mid a(X) = \gamma] \\
&= \text{MSE}_{\gamma}(\hat{\mathbb{P}}).
\end{aligned}
$$

Similarly, we have $\text{MSE}_{\gamma}(\hat{\mathbb{P}}_{AAR}) \leq \text{MSE}_{\gamma}(\hat{\mathbb{P}}_{TS})$, and therefore equation 18 holds.

Combining with equation 16 and equation 17, we have

$$CE_{\gamma}(\hat{\mathbb{P}}_{AAR}) \leq \min\{CE_{\gamma}(\hat{\mathbb{P}}_{TS}), CE_{\gamma}(\hat{\mathbb{P}})\}.$$

Table 4: **Conformal Calibration with Atypicality-Awareness.**

| Model | Dataset | Input Atypicality Group | Confidence Group | APS Coverage | APS SetSize | AA-APS Coverage | AA-APS SetSize | RAPS Coverage | RAPS SetSize | AA-RAPS Coverage | AA-RAPS SetSize |
|---|---|---|---|---|---|---|---|---|---|---|---|
| ResNet152 | ImageNet | 1 | 1 | 0.982 ± 0.002 | 51.192 ± 1.570 | 0.970 ± 0.003 | 32.694 ± 3.517 | 0.951 ± 0.004 | 7.168 ± 0.127 | 0.963 ± 0.004 | 14.408 ± 2.275 |
| | | | 2 | 0.984 ± 0.001 | 80.425 ± 2.506 | 0.961 ± 0.002 | 39.752 ± 1.670 | 0.940 ± 0.003 | 7.771 ± 0.180 | 0.966 ± 0.004 | 15.713 ± 0.977 |
| | | | 3 | 0.980 ± 0.001 | 114.526 ± 1.293 | 0.955 ± 0.004 | 55.887 ± 2.830 | 0.885 ± 0.005 | 8.485 ± 0.217 | 0.952 ± 0.004 | 25.191 ± 1.349 |
| | | | 4 | 0.979 ± 0.001 | 118.819 ± 1.787 | 0.949 ± 0.002 | 60.998 ± 1.293 | 0.833 ± 0.005 | 8.677 ± 0.235 | 0.949 ± 0.002 | 33.256 ± 0.812 |
| | | | 5 | 0.975 ± 0.001 | 111.696 ± 1.090 | 0.953 ± 0.002 | 66.884 ± 1.347 | 0.807 ± 0.006 | 8.801 ± 0.242 | 0.948 ± 0.002 | 42.109 ± 0.829 |
| | | | 6 | 0.958 ± 0.002 | 90.283 ± 1.460 | 0.949 ± 0.002 | 77.404 ± 1.682 | 0.777 ± 0.005 | 8.565 ± 0.229 | 0.946 ± 0.003 | 52.214 ± 1.401 |
| | | 2 | 1 | 0.961 ± 0.004 | 16.767 ± 0.389 | 0.953 ± 0.004 | 14.533 ± 1.036 | 0.977 ± 0.002 | 5.279 ± 0.035 | 0.954 ± 0.005 | 3.888 ± 0.102 |
| | | | 2 | 0.964 ± 0.002 | 18.359 ± 0.530 | 0.951 ± 0.003 | 12.902 ± 0.814 | 0.974 ± 0.001 | 5.801 ± 0.046 | 0.950 ± 0.003 | 4.484 ± 0.079 |
| | | | 3 | 0.961 ± 0.001 | 21.409 ± 0.413 | 0.948 ± 0.002 | 14.636 ± 0.318 | 0.966 ± 0.002 | 5.964 ± 0.051 | 0.952 ± 0.003 | 5.172 ± 0.112 |
| | | | 4 | 0.964 ± 0.002 | 23.162 ± 0.321 | 0.956 ± 0.003 | 20.246 ± 0.634 | 0.966 ± 0.002 | 6.253 ± 0.081 | 0.962 ± 0.004 | 6.485 ± 0.112 |
| | | | 5 | 0.953 ± 0.002 | 24.833 ± 0.486 | 0.952 ± 0.003 | 25.470 ± 0.829 | 0.938 ± 0.002 | 6.331 ± 0.092 | 0.953 ± 0.003 | 9.446 ± 0.332 |
| | | | 6 | 0.929 ± 0.003 | 21.124 ± 0.290 | 0.943 ± 0.004 | 32.341 ± 1.343 | 0.909 ± 0.003 | 6.409 ± 0.106 | 0.952 ± 0.002 | 16.970 ± 0.828 |
| | | 3 | 1 | 0.957 ± 0.002 | 6.843 ± 0.249 | 0.954 ± 0.004 | 5.860 ± 0.354 | 0.984 ± 0.001 | 4.839 ± 0.046 | 0.951 ± 0.004 | 3.015 ± 0.098 |
| | | | 2 | 0.963 ± 0.003 | 6.654 ± 0.202 | 0.965 ± 0.003 | 6.689 ± 0.406 | 0.986 ± 0.001 | 5.274 ± 0.044 | 0.966 ± 0.003 | 3.494 ± 0.071 |
| | | | 3 | 0.951 ± 0.002 | 7.609 ± 0.195 | 0.955 ± 0.003 | 8.542 ± 0.472 | 0.968 ± 0.001 | 5.435 ± 0.042 | 0.951 ± 0.003 | 4.052 ± 0.097 |
| | | | 4 | 0.938 ± 0.002 | 8.415 ± 0.205 | 0.955 ± 0.002 | 12.476 ± 0.349 | 0.961 ± 0.002 | 5.728 ± 0.048 | 0.951 ± 0.002 | 5.085 ± 0.085 |
| | | | 5 | 0.945 ± 0.002 | 8.500 ± 0.204 | 0.955 ± 0.003 | 12.539 ± 0.815 | 0.964 ± 0.002 | 5.724 ± 0.050 | 0.956 ± 0.003 | 5.531 ± 0.154 |
| | | | 6 | 0.932 ± 0.004 | 8.009 ± 0.199 | 0.953 ± 0.004 | 13.942 ± 0.657 | 0.944 ± 0.003 | 5.744 ± 0.048 | 0.953 ± 0.003 | 7.718 ± 0.745 |
| | | 4 | 1 | 0.958 ± 0.001 | 1.480 ± 0.034 | 0.962 ± 0.003 | 1.828 ± 0.131 | 0.985 ± 0.001 | 3.810 ± 0.132 | 0.962 ± 0.003 | 1.613 ± 0.078 |
| | | | 2 | 0.953 ± 0.002 | 1.388 ± 0.021 | 0.958 ± 0.002 | 1.913 ± 0.118 | 0.982 ± 0.001 | 3.816 ± 0.141 | 0.957 ± 0.002 | 1.692 ± 0.070 |
| | | | 3 | 0.931 ± 0.002 | 1.451 ± 0.028 | 0.947 ± 0.003 | 2.699 ± 0.146 | 0.967 ± 0.002 | 3.920 ± 0.148 | 0.951 ± 0.003 | 2.237 ± 0.071 |
| | | | 4 | 0.935 ± 0.001 | 1.450 ± 0.020 | 0.958 ± 0.003 | 3.488 ± 0.224 | 0.976 ± 0.002 | 3.880 ± 0.128 | 0.957 ± 0.004 | 2.577 ± 0.117 |
| | | | 5 | 0.928 ± 0.002 | 1.424 ± 0.018 | 0.951 ± 0.003 | 5.173 ± 0.391 | 0.963 ± 0.002 | 4.160 ± 0.127 | 0.963 ± 0.002 | 3.185 ± 0.077 |
| | | | 6 | 0.902 ± 0.003 | 1.476 ± 0.028 | 0.958 ± 0.005 | 8.900 ± 0.726 | 0.955 ± 0.004 | 3.925 ± 0.130 | 0.959 ± 0.004 | 4.235 ± 0.163 |
| | | 5 | 1 | 0.988 ± 0.001 | 1.000 ± 0.000 | 0.988 ± 0.001 | 1.000 ± 0.000 | 0.989 ± 0.001 | 1.344 ± 0.116 | 0.988 ± 0.001 | 1.000 ± 0.000 |
| | | | 2 | 0.982 ± 0.001 | 1.000 ± 0.000 | 0.982 ± 0.001 | 1.000 ± 0.000 | 0.984 ± 0.001 | 1.363 ± 0.122 | 0.982 ± 0.001 | 1.000 ± 0.000 |
| | | | 3 | 0.976 ± 0.001 | 1.000 ± 0.000 | 0.976 ± 0.001 | 1.001 ± 0.001 | 0.980 ± 0.002 | 1.351 ± 0.122 | 0.976 ± 0.001 | 1.001 ± 0.001 |
| | | | 4 | 0.977 ± 0.001 | 1.000 ± 0.000 | 0.977 ± 0.001 | 1.020 ± 0.009 | 0.980 ± 0.001 | 1.363 ± 0.109 | 0.977 ± 0.001 | 1.004 ± 0.003 |
| | | | 5 | 0.965 ± 0.001 | 1.000 ± 0.000 | 0.965 ± 0.001 | 1.137 ± 0.055 | 0.968 ± 0.002 | 1.342 ± 0.115 | 0.965 ± 0.001 | 1.063 ± 0.028 |
| | | | 6 | 0.970 ± 0.002 | 1.000 ± 0.000 | 0.972 ± 0.003 | 1.607 ± 0.125 | 0.973 ± 0.003 | 1.339 ± 0.107 | 0.973 ± 0.003 | 1.539 ± 0.080 |
| | | 6 | 1 | 0.992 ± 0.000 | 1.000 ± 0.000 | 0.992 ± 0.000 | 1.000 ± 0.000 | 0.992 ± 0.000 | 1.000 ± 0.000 | 0.992 ± 0.000 | 1.000 ± 0.000 |
| | | | 2 | 0.989 ± 0.001 | 1.000 ± 0.000 | 0.989 ± 0.001 | 1.000 ± 0.000 | 0.989 ± 0.001 | 1.000 ± 0.000 | 0.989 ± 0.001 | 1.000 ± 0.000 |
| | | | 3 | 0.992 ± 0.000 | 1.000 ± 0.000 | 0.992 ± 0.000 | 1.000 ± 0.000 | 0.992 ± 0.000 | 1.000 ± 0.000 | 0.992 ± 0.000 | 1.000 ± 0.000 |
| | | | 4 | 0.995 ± 0.001 | 1.000 ± 0.000 | 0.995 ± 0.001 | 1.000 ± 0.000 | 0.995 ± 0.001 | 1.000 ± 0.000 | 0.995 ± 0.001 | 1.000 ± 0.000 |
| | | | 5 | 0.989 ± 0.001 | 1.000 ± 0.000 | 0.989 ± 0.001 | 1.000 ± 0.000 | 0.989 ± 0.001 | 1.000 ± 0.000 | 0.989 ± 0.001 | 1.000 ± 0.000 |
| | | | 6 | 0.991 ± 0.001 | 1.000 ± 0.000 | 0.991 ± 0.001 | 1.000 ± 0.000 | 0.991 ± 0.001 | 1.000 ± 0.000 | 0.991 ± 0.001 | 1.000 ± 0.000 |
| ResNet18 | ImageNet | 1 | 1 | 0.982 ± 0.001 | 167.265 ± 2.058 | 0.952 ± 0.004 | 79.307 ± 3.180 | 0.876 ± 0.008 | 20.623 ± 1.724 | 0.952 ± 0.004 | 67.925 ± 2.552 |
| | | | 2 | 0.986 ± 0.001 | 157.710 ± 1.132 | 0.956 ± 0.002 | 85.303 ± 1.015 | 0.848 ± 0.009 | 20.740 ± 1.745 | 0.958 ± 0.003 | 80.333 ± 3.054 |
| | | | 3 | 0.970 ± 0.001 | 147.730 ± 1.267 | 0.943 ± 0.003 | 95.805 ± 2.283 | 0.810 ± 0.009 | 20.689 ± 1.725 | 0.938 ± 0.003 | 91.786 ± 1.213 |
| | | | 4 | 0.963 ± 0.002 | 129.188 ± 1.035 | 0.954 ± 0.003 | 109.238 ± 1.837 | 0.815 ± 0.009 | 20.328 ± 1.592 | 0.953 ± 0.002 | 102.926 ± 2.651 |
| | | | 5 | 0.957 ± 0.002 | 111.271 ± 1.097 | 0.952 ± 0.002 | 103.457 ± 2.047 | 0.809 ± 0.011 | 19.932 ± 1.441 | 0.942 ± 0.004 | 91.695 ± 3.655 |
| | | | 6 | 0.941 ± 0.004 | 82.127 ± 1.167 | 0.951 ± 0.005 | 96.879 ± 4.165 | 0.828 ± 0.008 | 19.108 ± 1.138 | 0.950 ± 0.004 | 87.490 ± 3.277 |
| | | 2 | 1 | 0.972 ± 0.001 | 37.918 ± 1.031 | 0.958 ± 0.001 | 25.001 ± 0.687 | 0.978 ± 0.002 | 15.096 ± 0.135 | 0.959 ± 0.003 | 10.650 ± 0.322 |
| | | | 2 | 0.959 ± 0.002 | 42.117 ± 0.743 | 0.951 ± 0.004 | 31.357 ± 1.294 | 0.954 ± 0.002 | 15.879 ± 0.162 | 0.951 ± 0.004 | 16.748 ± 0.617 |
| | | | 3 | 0.964 ± 0.003 | 43.312 ± 0.868 | 0.957 ± 0.003 | 39.329 ± 1.425 | 0.951 ± 0.003 | 16.138 ± 0.225 | 0.960 ± 0.002 | 22.619 ± 1.125 |
| | | | 4 | 0.951 ± 0.003 | 41.803 ± 0.701 | 0.955 ± 0.003 | 45.617 ± 1.420 | 0.930 ± 0.002 | 16.357 ± 0.295 | 0.953 ± 0.003 | 31.513 ± 1.447 |
| | | | 5 | 0.945 ± 0.002 | 36.697 ± 0.482 | 0.957 ± 0.002 | 45.379 ± 1.101 | 0.935 ± 0.002 | 16.094 ± 0.249 | 0.961 ± 0.001 | 32.569 ± 0.974 |
| | | | 6 | 0.930 ± 0.003 | 27.148 ± 0.321 | 0.949 ± 0.005 | 43.307 ± 2.384 | 0.924 ± 0.003 | 15.730 ± 0.144 | 0.948 ± 0.005 | 28.152 ± 1.657 |
| | | 3 | 1 | 0.964 ± 0.001 | 15.809 ± 0.414 | 0.954 ± 0.002 | 13.080 ± 0.586 | 0.985 ± 0.002 | 13.208 ± 0.471 | 0.955 ± 0.002 | 6.864 ± 0.146 |
| | | | 2 | 0.960 ± 0.002 | 18.034 ± 0.515 | 0.956 ± 0.003 | 16.515 ± 0.618 | 0.978 ± 0.001 | 13.659 ± 0.394 | 0.955 ± 0.002 | 8.275 ± 0.221 |
| | | | 3 | 0.964 ± 0.002 | 19.231 ± 0.344 | 0.957 ± 0.003 | 17.463 ± 0.722 | 0.979 ± 0.001 | 13.901 ± 0.310 | 0.957 ± 0.003 | 8.794 ± 0.163 |
| | | | 4 | 0.950 ± 0.001 | 19.479 ± 0.207 | 0.952 ± 0.003 | 21.470 ± 0.744 | 0.974 ± 0.001 | 13.899 ± 0.304 | 0.955 ± 0.003 | 11.390 ± 0.313 |
| | | | 5 | 0.940 ± 0.002 | 17.801 ± 0.160 | 0.953 ± 0.003 | 22.066 ± 0.545 | 0.957 ± 0.002 | 14.036 ± 0.296 | 0.951 ± 0.005 | 12.743 ± 0.271 |
| | | | 6 | 0.931 ± 0.002 | 13.096 ± 0.153 | 0.949 ± 0.004 | 21.352 ± 1.080 | 0.956 ± 0.002 | 13.169 ± 0.394 | 0.949 ± 0.005 | 12.924 ± 0.663 |
| | | 4 | 1 | 0.964 ± 0.002 | 5.377 ± 0.082 | 0.957 ± 0.003 | 4.595 ± 0.226 | 0.990 ± 0.002 | 11.670 ± 0.870 | 0.957 ± 0.004 | 3.823 ± 0.144 |
| | | | 2 | 0.958 ± 0.002 | 6.371 ± 0.130 | 0.954 ± 0.003 | 5.887 ± 0.275 | 0.986 ± 0.003 | 11.954 ± 0.767 | 0.953 ± 0.003 | 4.648 ± 0.106 |
| | | | 3 | 0.946 ± 0.002 | 5.810 ± 0.109 | 0.953 ± 0.002 | 7.575 ± 0.321 | 0.984 ± 0.003 | 12.145 ± 0.751 | 0.954 ± 0.003 | 5.364 ± 0.157 |
| | | | 4 | 0.940 ± 0.001 | 6.698 ± 0.159 | 0.950 ± 0.003 | 9.025 ± 0.498 | 0.980 ± 0.002 | 12.432 ± 0.700 | 0.950 ± 0.004 | 6.110 ± 0.225 |
| | | | 5 | 0.926 ± 0.003 | 5.817 ± 0.102 | 0.949 ± 0.004 | 10.234 ± 0.682 | 0.977 ± 0.003 | 12.325 ± 0.750 | 0.950 ± 0.004 | 6.925 ± 0.198 |
| | | | 6 | 0.915 ± 0.002 | 4.948 ± 0.098 | 0.949 ± 0.003 | 11.054 ± 0.601 | 0.968 ± 0.004 | 11.478 ± 0.813 | 0.948 ± 0.003 | 7.368 ± 0.183 |
| | | 5 | 1 | 0.963 ± 0.002 | 1.193 ± 0.008 | 0.968 ± 0.002 | 1.497 ± 0.052 | 0.989 ± 0.002 | 8.559 ± 1.391 | 0.967 ± 0.002 | 1.413 ± 0.041 |
| | | | 2 | 0.959 ± 0.002 | 1.232 ± 0.014 | 0.963 ± 0.002 | 1.456 ± 0.063 | 0.984 ± 0.003 | 8.755 ± 1.352 | 0.962 ± 0.002 | 1.485 ± 0.057 |
| | | | 3 | 0.957 ± 0.001 | 1.243 ± 0.012 | 0.961 ± 0.002 | 1.693 ± 0.085 | 0.986 ± 0.003 | 8.947 ± 1.344 | 0.961 ± 0.002 | 1.637 ± 0.071 |
| | | | 4 | 0.945 ± 0.002 | 1.264 ± 0.013 | 0.959 ± 0.002 | 2.492 ± 0.131 | 0.985 ± 0.003 | 8.933 ± 1.325 | 0.959 ± 0.002 | 2.309 ± 0.082 |
| | | | 5 | 0.943 ± 0.002 | 1.234 ± 0.013 | 0.953 ± 0.002 | 2.770 ± 0.345 | 0.976 ± 0.004 | 8.913 ± 1.365 | 0.954 ± 0.002 | 2.540 ± 0.195 |
| | | | 6 | 0.932 ± 0.002 | 1.214 ± 0.010 | 0.961 ± 0.002 | 3.875 ± 0.249 | 0.976 ± 0.004 | 8.459 ± 1.351 | 0.963 ± 0.003 | 3.635 ± 0.192 |
| | | 6 | 1 | 0.991 ± 0.001 | 1.000 ± 0.000 | 0.991 ± 0.001 | 1.000 ± 0.000 | 0.903 ± 0.001 | 4.540 ± 1.729 | 0.991 ± 0.001 | 1.000 ± 0.000 |
| | | | 2 | 0.986 ± 0.001 | 1.000 ± 0.000 | 0.986 ± 0.001 | 1.000 ± 0.000 | 0.990 ± 0.002 | 4.395 ± 1.738 | 0.986 ± 0.001 | 1.000 ± 0.000 |
| | | | 3 | 0.988 ± 0.001 | 1.000 ± 0.000 | 0.988 ± 0.001 | 1.000 ± 0.000 | 0.992 ± 0.001 | 4.449 ± 1.739 | 0.988 ± 0.001 | 1.000 ± 0.000 |
| | | | 4 | 0.988 ± 0.001 | 1.000 ± 0.000 | 0.988 ± 0.001 | 1.000 ± 0.000 | 0.991 ± 0.002 | 4.481 ± 1.729 | 0.988 ± 0.001 | 1.000 ± 0.000 |
| | | | 5 | 0.986 ± 0.001 | 1.000 ± 0.000 | 0.986 ± 0.001 | 1.000 ± 0.000 | 0.990 ± 0.002 | 4.390 ± 1.741 | 0.986 ± 0.001 | 1.000 ± 0.000 |
| | | | 6 | 0.981 ± 0.001 | 1.000 ± 0.000 | 0.981 ± 0.001 | 1.003 ± 0.003 | 0.987 ± 0.002 | 4.146 ± 1.767 | 0.981 ± 0.001 | 1.009 ± 0.007 |
| ResNet50 | ImageNet | 1 | 1 | 0.980 ± 0.002 | 80.190 ± 1.594 | 0.955 ± 0.004 | 50.203 ± 2.778 | 0.945 ± 0.002 | 9.874 ± 0.169 | 0.964 ± 0.002 | 16.099 ± 0.593 |
| | | | 2 | 0.986 ± 0.002 | 118.839 ± 1.737 | 0.955 ± 0.003 | 52.066 ± 2.094 | 0.896 ± 0.003 | 10.681 ± 0.209 | 0.950 ± 0.003 | 25.545 ± 0.845 |
| | | | 3 | 0.979 ± 0.001 | 134.702 ± 0.697 | 0.951 ± 0.003 | 60.743 ± 1.012 | 0.848 ± 0.004 | 11.056 ± 0.228 | 0.940 ± 0.002 | 35.518 ± 0.870 |
| | | | 4 | 0.976 ± 0.001 | 126.892 ± 0.939 | 0.952 ± 0.003 | 79.627 ± 2.106 | 0.835 ± 0.004 | 11.079 ± 0.230 | 0.953 ± 0.004 | 51.339 ± 1.445 |
| | | | 5 | 0.964 ± 0.001 | 114.559 ± 0.975 | 0.954 ± 0.003 | 93.609 ± 3.277 | 0.797 ± 0.005 | 11.011 ± 0.232 | 0.942 ± 0.003 | 63.441 ± 1.422 |
| | | | 6 | 0.949 ± 0.002 | 88.034 ± 0.799 | 0.950 ± 0.003 | 88.418 ± 1.898 | 0.806 ± 0.005 | 10.744 ± 0.221 | 0.957 ± 0.002 | 62.049 ± 1.862 |
| | | 2 | 1 | 0.961 ± 0.002 | 17.461 ± 0.533 | 0.952 ± 0.004 | 14.757 ± 0.785 | 0.976 ± 0.002 | 7.080 ± 0.116 | 0.956 ± 0.004 | 5.205 ± 0.127 |
| | | | 2 | 0.967 ± 0.002 | 24.896 ± 0.732 | 0.958 ± 0.002 | 20.892 ± 1.441 | 0.965 ± 0.002 | 8.013 ± 0.106 | 0.958 ± 0.002 | 6.879 ± 0.191 |
| | | | 3 | 0.961 ± 0.001 | 29.101 ± 0.759 | 0.947 ± 0.003 | 21.754 ± 1.123 | 0.963 ± 0.001 | 8.134 ± 0.126 | 0.956 ± 0.003 | 8.773 ± 0.321 |
| | | | 4 | 0.958 ± 0.002 | 30.078 ± 0.379 | 0.956 ± 0.003 | 28.199 ± 0.951 | 0.946 ± 0.002 | 8.398 ± 0.129 | 0.951 ± 0.003 | 11.264 ± 0.347 |
| | | | 5 | 0.944 ± 0.001 | 29.401 ± 0.357 | 0.946 ± 0.002 | 32.362 ± 1.180 | 0.937 ± 0.003 | 8.559 ± 0.143 | 0.952 ± 0.002 | 15.040 ± 0.547 |
| | | | 6 | 0.939 ± 0.002 | 22.100 ± 0.488 | 0.954 ± 0.004 | 33.527 ± 1.955 | 0.922 ± 0.003 | 8.269 ± 0.135 | 0.955 ± 0.001 | 18.645 ± 0.283 |
| | | 3 | 1 | 0.958 ± 0.002 | 9.385 ± 0.202 | 0.953 ± 0.002 | 7.633 ± 0.346 | 0.988 ± 0.001 | 6.471 ± 0.129 | 0.955 ± 0.002 | 3.820 ± 0.076 |
| | | | 2 | 0.951 ± 0.003 | 9.619 ± 0.194 | 0.949 ± 0.005 | 8.739 ± 0.516 | 0.977 ± 0.002 | 6.919 ± 0.118 | 0.950 ± 0.004 | 4.648 ± 0.082 |
| | | | 3 | 0.957 ± 0.002 | 11.449 ± 0.274 | 0.957 ± 0.003 | 10.853 ± 0.446 | 0.975 ± 0.001 | 7.163 ± 0.112 | 0.954 ± 0.004 | 5.130 ± 0.102 |
| | | | 4 | 0.953 ± 0.001 | 10.809 ± 0.223 | 0.951 ± 0.002 | 10.546 ± 0.396 | 0.972 ± 0.001 | 7.600 ± 0.113 | 0.953 ± 0.003 | 5.872 ± 0.235 |
| | | | 5 | 0.947 ± 0.002 | 11.507 ± 0.228 | 0.951 ± 0.003 | 14.754 ± 0.650 | 0.966 ± 0.003 | 7.414 ± 0.114 | 0.953 ± 0.004 | 6.874 ± 0.157 |
| | | | 6 | 0.920 ± 0.003 | 9.150 ± 0.188 | 0.951 ± 0.005 | 20.041 ± 1.624 | 0.944 ± 0.002 | 7.143 ± 0.101 | 0.950 ± 0.002 | 9.505 ± 0.576 |
| | | 4 | 1 | 0.955 ± 0.002 | 1.739 ± 0.025 | 0.953 ± 0.002 | 1.667 ± 0.070 | 0.985 ± 0.002 | 5.088 ± 0.196 | 0.954 ± 0.002 | 1.574 ± 0.038 |
| | | | 2 | 0.948 ± 0.002 | 1.958 ± 0.016 | 0.950 ± 0.002 | 1.964 ± 0.083 | 0.983 ± 0.001 | 5.285 ± 0.189 | 0.951 ± 0.003 | 1.845 ± 0.061 |
| | | | 3 | 0.936 ± 0.003 | 2.082 ± 0.034 | 0.955 ± 0.002 | 4.255 ± 0.202 | 0.976 ± 0.002 | 5.495 ± 0.164 | 0.957 ± 0.003 | 3.423 ± 0.123 |
| | | | 4 | 0.936 ± 0.003 | 2.173 ± 0.045 | 0.955 ± 0.003 | 4.508 ± 0.303 | 0.973 ± 0.002 | 5.573 ± 0.170 | 0.952 ± 0.002 | 3.013 ± 0.110 |
| | | | 5 | 0.928 ± 0.002 | 2.071 ± 0.041 | 0.953 ± 0.003 | 8.483 ± 0.789 | 0.968 ± 0.002 | 5.720 ± 0.167 | 0.951 ± 0.002 | 4.338 ± 0.138 |
| | | | 6 | 0.895 ± 0.004 | 1.932 ± 0.026 | 0.950 ± 0.005 | 10.274 ± 1.147 | 0.950 ± 0.003 | 5.379 ± 0.151 | 0.948 ± 0.004 | 5.270 ± 0.220 |
| | | 5 | 1 | 0.979 ± 0.001 | 1.000 ± 0.000 | 0.979 ± 0.001 | 1.001 ± 0.001 | 0.985 ± 0.001 | 1.918 ± 0.222 | 0.979 ± 0.001 | 1.000 ± 0.000 |
| | | | 2 | 0.982 ± 0.001 | 1.000 ± 0.000 | 0.982 ± 0.001 | 1.015 ± 0.008 | 0.985 ± 0.001 | 1.880 ± 0.221 | 0.982 ± 0.001 | 1.019 ± 0.007 |
| | | | 3 | 0.964 ± 0.002 | 1.000 ± 0.000 | 0.964 ± 0.002 | 1.126 ± 0.028 | 0.971 ± 0.002 | 1.977 ± 0.222 | 0.965 ± 0.002 | 1.178 ± 0.045 |
| | | | 4 | 0.972 ± 0.002 | 1.000 ± 0.000 | 0.973 ± 0.002 | 1.186 ± 0.047 | 0.978 ± 0.002 | 1.958 ± 0.205 | 0.973 ± 0.002 | 1.156 ± 0.050 |
| | | | 5 | 0.968 ± 0.002 | 1.000 ± 0.000 | 0.971 ± 0.002 | 1.417 ± 0.102 | 0.974 ± 0.002 | 1.911 ± 0.222 | 0.971 ± 0.002 | 1.293 ± 0.067 |
| | | | 6 | 0.946 ± 0.002 | 1.000 ± 0.000 | 0.957 ± 0.002 | 1.842 ± 0.143 | 0.956 ± 0.003 | 1.833 ± 0.193 | 0.957 ± 0.002 | 1.763 ± 0.112 |
| | | 6 | 1 | 0.991 ± 0.001 | 1.000 ± 0.000 | 0.991 ± 0.001 | 1.000 ± 0.000 | 0.991 ± 0.001 | 1.000 ± 0.000 | 0.991 ± 0.001 | 1.000 ± 0.000 |
| | | | 2 | 0.991 ± 0.001 | 1.000 ± 0.000 | 0.991 ± 0.001 | 1.000 ± 0.000 | 0.991 ± 0.001 | 1.000 ± 0.000 | 0.991 ± 0.001 | 1.000 ± 0.000 |
| | | | 3 | 0.994 ± 0.001 | 1.000 ± 0.000 | 0.994 ± 0.001 | 1.000 ± 0.000 | 0.994 ± 0.001 | 1.000 ± 0.000 | 0.994 ± 0.001 | 1.000 ± 0.000 |
| | | | 4 | 0.992 ± 0.001 | 1.000 ± 0.000 | 0.992 ± 0.001 | 1.000 ± 0.000 | 0.992 ± 0.001 | 1.000 ± 0.000 | 0.992 ± 0.001 | 1.000 ± 0.000 |
| | | | 5 | 0.992 ± 0.001 | 1.000 ± 0.000 | 0.992 ± 0.001 | 1.000 ± 0.000 | 0.992 ± 0.001 | 1.000 ± 0.000 | 0.992 ± 0.001 | 1.000 ± 0.000 |
| | | | 6 | 0.987 ± 0.001 | 1.000 ± 0.000 | 0.987 ± 0.001 | 1.000 ± 0.000 | 0.987 ± 0.001 | 1.000 ± 0.000 | 0.987 ± 0.001 | 1.000 ± 0.000 |

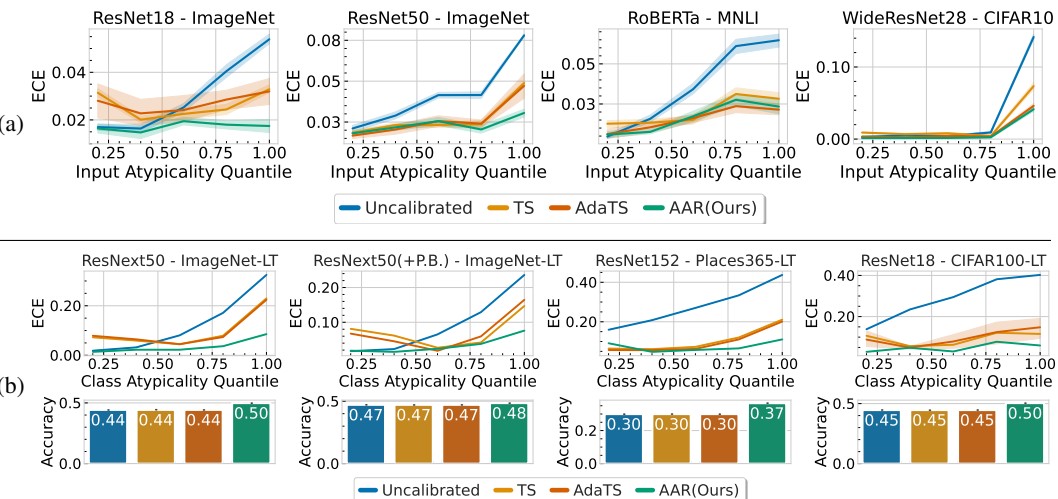

Figure 13: **Post-hoc Recalibration for Classification with 5-NN distance as an Atypicality Metric.** **(a) Balanced Supervised Classification:** Atypicality-Aware Recalibration improves the calibration of models trained with balanced datasets, across atypicality groups. **(b) Imbalanced Supervised Classification:** Atypicality-Aware Recalibration improves both the calibration across groups and the overall accuracy of models trained with imbalanced datasets with 5-nearest neighbors distance as an atypicality metric.

# H  Limitations

## H.1  Quantifying Atypicality

Since we do not have access to the true distribution of $\mathbb{P}(X)$, we estimate it through the model, e.g. using the embeddings. This means we are capturing the atypicality not solely with respect to the training distribution but also the model. It is possible that a model that does not fit the data well and produces low-quality atypicality estimates. *We would like to stress that our goal here is to show that even simple estimators can demonstrate significant benefits.* In general, we observe that our findings hold for large datasets and widely used models, and atypicality gives a semantically meaningful way to group data points qualitatively. Our findings suggest that we can unify the understanding and improve uncertainty quantification and recalibration methods with atypicality, however, practitioners should be careful about incorporating atypicality, as poor atypicality estimates can lead to worse performance.

## H.2  Subgroup Fairness Experiments

While the literature on algorithms to satisfy group fairness is rich [KGZ19, HJKRR18], here we wanted to give a case study with skin-lesion classification. Our goal was to provide further evidence that atypicality awareness could improve fairness algorithms. In this domain, there is more verification to do to better characterize how and when atypicality helps, thus, to better understand which subgroups could benefit more from atypicality-aware algorithms and whether these findings apply generally in the literature.

## H.3  Theoretical Analysis

Following the earlier work [BMWX21a, SC19], we analyzed the calibration behavior of well-specified logistic regression. However, our empirical findings suggest that the phenomena are much more broadly applicable. We suggest that future work can analyze the behavior in more general settings to better understand the dynamics.

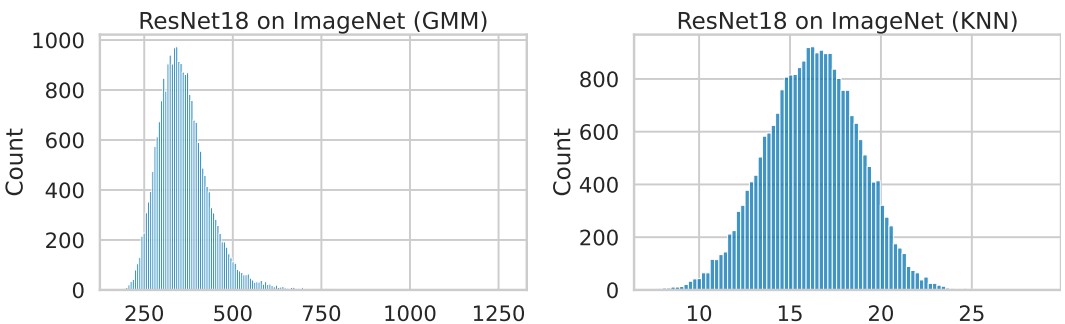

Figure 14: **Distribution of Input Atypicality.** Here, we give the distribution of Atypicality for ResNet18 on ImageNet, using GMM and KNN methods.

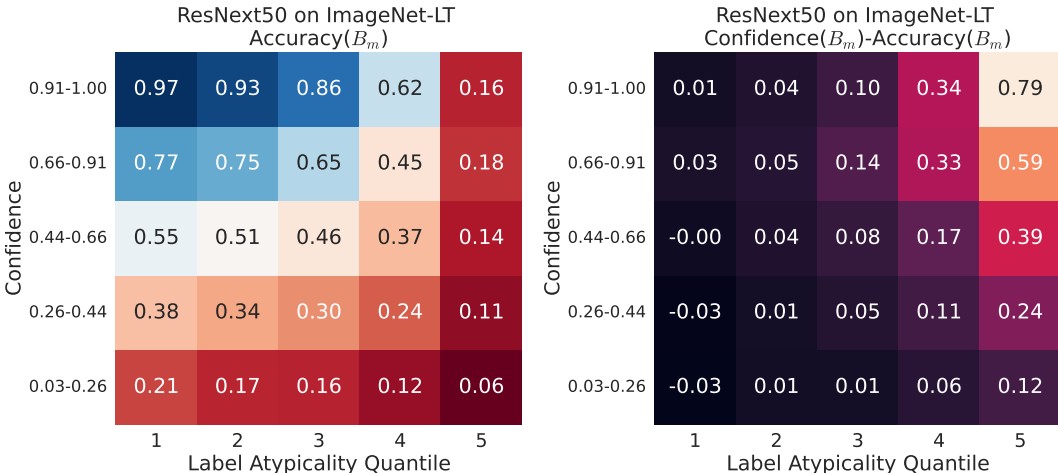

Figure 15: **Label Atypicality and Confidence.** Here, the x-axis reflects the label atypicality quantile, and the y-axis indicates confidence. The coloring for the figure on the left indicates the accuracy within a bin, and the figure on the right has the difference between confidence and accuracy within a bin. Similar to Figure 6, we observe that atypical examples have lower accuracy, and predictions are more overconfident.

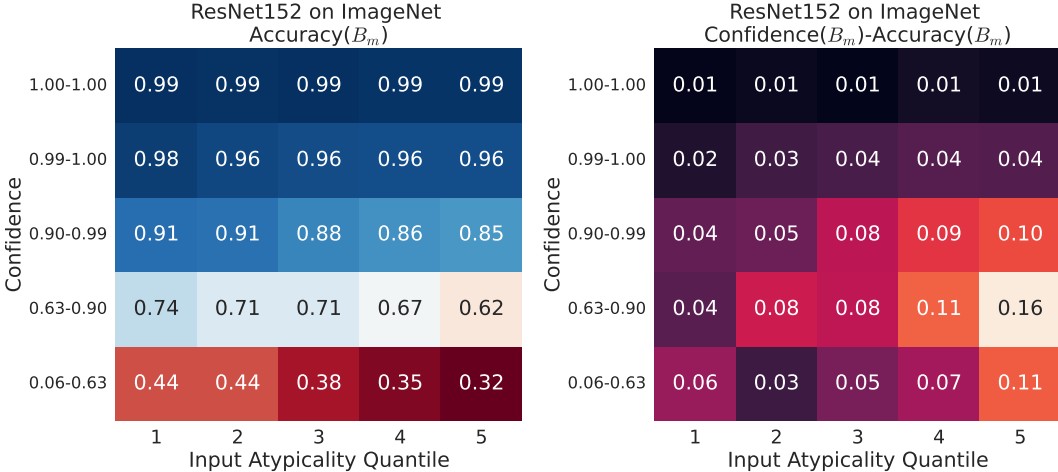

Figure 16: **Input Atypicality and Confidence for ResNet152.** We provide the input atypicality for ResNet152, in the same structure as Figure 15.

