# OpenReview forum: "Beyond Confidence: Reliable Models Should Also Consider Atypicality"
_NeurIPS.cc/2023/Conference — NeurIPS 2023 poster_

### Official Review · Reviewer_7E3C · 2023-06-13

**Soundness:** 3 good
**Presentation:** 2 fair
**Contribution:** 2 fair
**Rating:** 5
**Confidence:** 5

**Summary:**

This paper focuses on the question of uncertainty quantification in classification by introducing the atypicality during recalibration. This paper exhibits a highly explicit motivation, yet it suffers from certain deficiencies in definitions and errors therein, and I harbor reservations regarding the soundness of the approach posited in Sec. 4.2 Atypicality-Aware Recalibration.


**Strengths:**

This paper evinces a highly explicit motivation and lucidly illustrates the concept of atypicality posited therein through the use of schematic diagrams.

The paper presents copious experimental results and supplements them with additional experiments in the appendix, which serves to facilitate the reader's comprehension of the experimental outcomes.


**Weaknesses:**

The presence of errors and deficiencies in definitions within the article renders this work somewhat unreliable.

Although the article contains numerous derivations, the critical section, namely Sec. 4.2, lacks theoretical analysis. The author fails to analyze the validity and necessity of Eq. 2, and whether there exist alternative methods to the approach posited therein, or whether it is indeed necessary. These issues require requisite analysis.

**Questions:**


Q1: In Fig.~1, it is unclear whether the example presented therein is a genuine instance from the experiment. If it is, then it is puzzling why specific values for atypicality and confidence are not provided, and instead, actual and predicted labels are employed.

Q2: There are errors in definition of accuracy(Bm), what's worse, I can't find the formal defination of Bm.

Q3: RMSE, which is used as a metric in the paper, but I can't find the definition of it. To my understanding, RMSE is a metric for regression rather than calibration.

Q4: The necessity of Eq. 2, and whether there exist alternative methods to the approach posited therein, or whether it is indeed necessary. I could understand Sec. 4.1, and the natural selution is group-wise TS calibration, so more detailed analysis is required.

Q5: Atypicality for LLMs. The defination in line 124 a(x) is a distribution, it seems different to the defination 2.1.

Q6: How is the expected calibration error (ECE) computed on large-scale models? Is it obtained by taking the softmax of all possible values along the final dimension, or by normalizing the top-p values as confidence scores?

Note: Use "Line xx" to describe the Eqs in the text is cumbersome and detracts from the readability of the communication.

---

> ### Author Rebuttal · Authors · 2023-08-09
>
> Dear Reviewer 7E3C,
>
> Thank you very much for your detailed review and your kindness! We appreciate the time you took to review, and we are really excited that you find our motivation clear and supported with copious experimental evidence!
>
> Below are the responses to your questions:
>
> ### Response to Questions / Comments
>
> - Q1. The instances are indeed from the ImageNet-R dataset. We sampled those points according to the atypicality and confidence values and their quantiles. In the revision, we will add the quantile values to Figure 1. Thank you for this suggestion.
>
> - Q2. Thank you for pointing this out, we further clarified the meaning of $B_{m}$ in Appendix D.1. Just noting here too: $B_{m}$ here denotes the set of the data indices where the confidence of the prediction for the sample falls into the interval $(\frac{m-1}{M}, \frac{m}{M}]$ where $M$ is the number of bins. For instance, if we have 10 bins, $B_{1}$ contains samples with confidence between $(0, 0.1]$.
>
> - Q3. The definition is Appendix D.1 Equation 8. RMSCE is simply the root-mean-squared version of the ECE metric. This metric is based on the earlier works e.g. the cited paper [HMD18].
>
> - Q4. Thank you for raising this point. Group-conditional calibration is possible when there is a natural definition of a group (as in the Fairness Experiments presented in Figure 4). However, when we do not have the definition of a group, it is unclear how many groups to form, according to what threshold, etc. In practice, we would face difficulties in picking the number of groups, finding thresholds to form groups, and justifying this choice. Supported by the parametric relation we observe in Appendix Figure 10 and therein, we follow the proposed methodology to overcome such practical hurdles. Indeed, Figure 10 suggests that there is a monotonic relationship between atypicality and temperature that could be captured by the simple parametric form that we have that requires only 3 parameters.
>
> - Q5. Thank you for raising this question, we believe it is important to clarify this. Intuitively, our atypicality measure aims to quantify how well-represented an input or a class is in the training data. $a(X)$ is a notion to evaluate whether an input is well-represented. The key idea is that a larger atypicality value indicates $X$ is not well-represented in the patterns seen during training.  $a(X)$ could be implemented in a class-conditional way by looking at $P(X|Y)$ for each class $Y$, or in a marginal way by looking at the overall $P(X)$, and we adopt the one that is more practical to use. For LLMs, it is unclear how one would quantify $P(X|Y)$, however, the model itself is an estimator of $P(X)$, making this notion of typicality readily available. This quantity still informs us about whether the input prompt is well-represented w.r.t. the training data. We will also add further discussion on this to the Appendix for further clarification.
>
> - Q6: We use the post-softmax probability for the most probable class as the confidence, and compute ECE using this quantity. This is indicated in Equation 7, but we will also add further clarification on this to the Appendix.
>
> Thank you so much for all your questions and comments, we really appreciate your time and kindness! Please let us know if there are any further questions that you may have.

---

> > ### Author Response · Authors · 2023-08-14
> > **We would love to hear from you!**
> >
> > Dear Reviewer 7E3C,
> >
> > Once again, we really appreciate your detailed review. As we are in the middle stage of the discussion period, we hope you find our responses useful, and we would like to ask if the questions you raised have been addressed. We would love to engage with you further if there are any remaining points.
> >
> > We understand that the discussion period is short, and we sincerely appreciate your time and help!

---

> > > ### Comment · Reviewer_7E3C · 2023-08-15
> > >
> > > I appreciate the response. I am happy to maintain my score.

---

> > > > ### Author Response · Authors · 2023-08-15
> > > > **Thank you!**
> > > >
> > > > Dear Reviewer 7E3C,
> > > >
> > > > Thank you for your response! Please do let us know if you have have any further questions about the work.

---

### Official Review · Reviewer_vUKB · 2023-06-13

**Soundness:** 2 fair
**Presentation:** 3 good
**Contribution:** 2 fair
**Rating:** 7
**Confidence:** 1

**Summary:**

This paper is addressing the problem of atypicality in data, and how this impact performance and confidence.

**Strengths:**

Good perspective on the needs to consider atypicality in data
Good presentation
Good review on the links between atypicality in data and performance with confidence

**Weaknesses:**

N/A

**Questions:**

N/a

---

> ### Author Rebuttal · Authors · 2023-08-09
>
> Dear Reviewer vUKB,
>
> Thank you for your kind remark. Let us know if you have any questions or comments.

---

### Official Review · Reviewer_AQHi · 2023-07-04

**Soundness:** 3 good
**Presentation:** 1 poor
**Contribution:** 2 fair
**Rating:** 5
**Confidence:** 4

**Summary:**

This paper proposes a series of "atypicality" measures to be used in complement to the more popular uncertainty metrics.
The authors defined input and class atypicality for both classical classification tasks as well as NLG.
Such atypicality measures are then combined with temperature scaling to improve calibration quality, as well as prediction sets (RAPS).

**Strengths:**

1. The problem is definitely important, and this paper touches a lot of areas where the particular measures of "atypicality" could help.
2. Linking atypicality to conformal prediction is good. (I'd suggest using "prediction set" as opposed to "uncertainty set" though). This seems to lead to good results in Figure 5 as well.
3. AAR seems to improve ECE over the baselines significantly.
4. I actually think this paper has a lot of interesting ideas, such as Theorem 3.1 (but somehow constrained by the space and many are not fully developed.)

**Weaknesses:**

1. I really think it is inappropriate to make the dichotomy of "Atypicality" vs "Confidence/Uncertainty". In fact, I would argue the various forms of atypicality in this paper are just a bunch of confidence measures. This paper actually also proposes several different variants of "atypicality", and in the uncertainty space, for example, monte-carlo dropout variance and entropy are both used to estimate uncertainty. Thus, I'm not sure what makes "atypicality" a separate concept.

2. In my opinion, this paper is proposing another calibration method, but puts everything in very different languages.
It is also missing a lot of calibration baselines.
In fact, recent works rarely focus on confidence calibration anymore, but more on full calibrtion [[1]](https://proceedings.neurips.cc/paper/2019/hash/8ca01ea920679a0fe3728441494041b9-Abstract.html), [[2]](http://proceedings.mlr.press/v89/vaicenavicius19a/vaicenavicius19a.pdf), [[3]](https://proceedings.neurips.cc/paper/2019/file/1c336b8080f82bcc2cd2499b4c57261d-Paper.pdf), [[4]](https://openreview.net/forum?id=p_jIy5QFB7).
In fact, [4] also used density estimates to calibrate classifiers much like L115-L117 and Definition 2.1
I also think the "Recalibration" section in Section 6 needs a re-write, as it's uncommon to put conformal prediction in this context: Yes, conformal prediction could be considered as calibrating the distribution/p-value, but when we use ECE to measure model calibration, conformal prediction is just out of context.

3. Related to 2, I think the improvement of RAPS/APS would benefit mostly from full calibration as opposed to confidence calibration. The lack of such experiment is a limitation.

4. While the improvement of LLM performance is good, I don't see the connection between the main point of the paper and such experiments. See Questions as well.

5. Presentation is not clear, probably due to the attempt to jam in too much content. For example, L122-126 should be expanded as it is a very different thing in my opinion.

**Questions:**

1. Related to W4, why is atypicality defined so differently for NLP tasks? The intro keeps suggesting that the input is atypical wrt a class (e.g. Figure 1). In Definition 2.1, for example, it is also conditioned on $Y=y$. However, L124 uses a completely different definition. In fact, even L125 ($a_Y(y)$) has a completely different meaning than Definition 2.2.
2. Intuitively, what's the rationale of assigning temperatures basing on atypicality? It seems like typical samples could share a temperature, but for atypical ones, it's not like atypical samples themselves are similar.
3. Are the theoretical guarantees still maintained after the modifications to RAPS?
4. Is Theorem 3.1 somewhat mechanical? That is, is this just due to large variance and the fact that $u-\mathbb{P}(Y=1|\hat{\mathbb{P}}_1(X) = u)$ tends to mean reverse?


**Limitations:**

Mostly discussed in weaknesses.

---

> ### Author Rebuttal · Authors · 2023-08-09
>
> Dear Reviewer AQHi,
>
> Thank you very much for your detailed review. We really appreciate that you find the problem important and could be helpful in practice; we share the same ideas! We are also very happy to hear that you find a lot of interesting ideas in the paper, thank you very much for your kindness!
>
> Below are our responses to your questions and comments:
>
> ### Responses to Questions / Comments
>
> - Atypicality vs Confidence: Confidence and atypicality are correlated quantities that are clearly not independent, so we agree that there is not a dichotomy per se. Our main message is to highlight if and when having the specific notions that we use could provide utility. As an illustration, let us think about the Bayes Rule: $P(Y|X) = \frac{P(X|Y)P(Y)}{\sum_{Y’ \in \mathcal{Y}} P(X | Y’)P(Y’)}$. For example, let us have two inputs $X_{1}, X_{2}$ where $X_{1}$ is an out-of-distribution example that is not from any of the classes in the problem, and we have $\forall Y’ \, P(X_{1} | Y’) < \epsilon$ for some $\epsilon$. Similarly, $\exists \tilde{Y}: P(X_{2} | \tilde{Y}) \gg \epsilon$ and this input is ‘in-distribution’. Looking at the Bayes Rule, we can have the same confidence distribution for the two points $(P(Y|X_{1}) = P(Y|X_{2}))$ as long as the likelihood ratios are the same, where $X_{1}$ is an OOD point, whereas $X_{2}$ is potentially from one of the classes. In other words, an atypical input $X_{1}$ that is not likely to be from any class can have the same confidence value as the typical input. Therefore, quantifying and accounting for both confidence and atypicality can provide value and in our study, we demonstrate the value in improving calibration and accuracy.
>
> - Comments on writing: We changed the term uncertainty set with prediction set in the paper. Similarly, we slightly modified the recalibration part of the related works to make the distinction for conformal and recalibration. We appreciate both of these suggestions.
>
> - Q1. Thank you for raising this question, we believe it is important to clarify this. Intuitively, our atypicality measure aims to quantify how well-represented an input or a class is in the training data. $a(X)$ is a notion to evaluate whether an input is well-represented. The key idea is that a larger atypicality value indicates $X$ is not well-represented in the patterns seen during training.  $a(X)$ could be implemented in a class-conditional way by looking at $P(X|Y)$ for each class $Y$, or in a marginal way by looking at the overall $P(X)$, and we adopt the one that is more practical to use. For LLMs, it is unclear how one would quantify $P(X|Y)$, however, the model itself is an estimator of $P(X)$, making this notion of typicality readily available. This quantity still informs us about whether the input prompt is well-represented w.r.t. the training data. We will also add further discussion on this to the Appendix for further clarification.
>
> - Q2. The intuition is that the model’s predictions are overconfident for atypical points. A larger temperature leads to the reduction of confidence and thus larger uncertainty, and we indeed observe larger temperatures for atypical samples. Overall, it is a way to refine the model’s confidence to reflect the higher uncertainty.
>
> - Q3. We have not theoretically analyzed the setting for prediction sets. Unfortunately, we do not believe such an analysis could be fit in this particular paper given the space constraints, but we will add further clarification on this under the ideas to future work. In particular, utilizing the proof techniques in [1], one can prove that for conformal prediction in the regression setting, an initialization $f_0$ with smaller MSE will have a shorter prediction interval after the post-processing of split conformal. Future work can extend this analysis to the classification setting.
>
> - Q4. Thank you for your comment. The larger variance of $\hat{\mathbb{P}}_1(X)$ for more atypical $X$ is indeed one cause of our result. While this can generally explain large calibration errors, it cannot specifically explain overconfidence. That is, it does provide a reason for why the sign of $u-\mathbb P(Y\mid \hat{\mathbb{P}}_1(X)=u)$ is positive. Our theorem provides a more specific analysis to characterize the overconfidence phenomenon.
>
> Thank you so much for all your questions and comments, we really appreciate your time! Please let us know if there are any further questions that you may have.
>
> [1] Lei, J., G’Sell, M., Rinaldo, A., Tibshirani, R. J., & Wasserman, L. (2018). Distribution-free predictive inference for regression. Journal of the American Statistical Association, 113(523), 1094-1111.

---

> > ### Author Response · Authors · 2023-08-14
> > **We would love to hear from you!**
> >
> > Dear Reviewer AQHi,
> >
> > Once again, we really appreciate your detailed review. As we are in the middle stage of the discussion period, we hope you find our responses useful, and we would like to ask if the questions you raised have been addressed. We would love to engage with you further if there are any remaining points.
> >
> > We understand that the discussion period is short, and we sincerely appreciate your time and help!

---

> > ### Comment · Reviewer_AQHi · 2023-08-18
> > **Thank you**
> >
> > Thank you for the responses. In general, I think the authors' responses conform to my original understanding. I don't think measuring the quantities currently called atypicality is bad, but I was just saying these are conceptually different concepts. For example, if P(X) in NLP is atypicality, then there should be a clear distinction between class-conditional atypicality vs non-class-conditional atypicality. This is currently not the main point of the paper.
> >
> > Q3: I think the point of RAPS is the coverage guarantee.
> > > an initialization
> >  with smaller MSE will have a shorter prediction interval after the post-processing of split conformal
> >
> > I agree, but I'm not sure why we need to mention RAPS if MSE is already the indicator. If there are experiments wrt to W3 then this might be more useful/convincing.
> >
> >
> > Q4: Could you explain why this is *not* related to mean-reversion? My original question was essentially that if u is random, condition on $u-\mathbb{P}(Y|\hat{\mathbb{P}}(X)=u)$ being large, it seems like its expectation will be smaller mechanically (because whatever "noise" will mean-revert)

---

> > > ### Author Response · Authors · 2023-08-19
> > > **Thank you!**
> > >
> > > Thank you for your response!
> > >
> > > - Q3: We believe without empirical evidence that we presented in the paper, it would not have been clear whether and how atypicality could be useful in conformal prediction (and how it would concretely relate to coverage). While we took a first step to demonstrate the utility of quantifying atypicality for conformal prediction, we agree that there is more empirical and theoretical investigation to do in the future work regarding full calibration and beyond, which we will note in our limitations section.
> > >
> > > - Q4: As one potential example, the results in Bai et al. show that under different activation functions, one can observe an underconfidence effect instead of overconfidence in a similar setting (e.g. see Equation 13 in Bai et al. for an activation function that induces underconfidence). Under such evidence, it is not clear to us whether there is or there is not more going on than general mean reversion. We are not broadly claiming it is not mean reversion but argue that this phenomenon is subject to further investigation.
> > >
> > > Once again, we appreciate that you find the ideas in the paper interesting, and you took the time to interact during this phase. Please let us know if you have any further questions, we are happy to follow up.

---

### Official Review · Reviewer_y4KB · 2023-07-07

**Soundness:** 3 good
**Presentation:** 3 good
**Contribution:** 4 excellent
**Rating:** 8
**Confidence:** 4

**Summary:**

The paper questions the reliance of probabilistic classifiers on the confidence score alone for measuring reliability. This is an important question that has not been asked that rigorously in the machine learning literature. In particular, there are two notions of uncertainty: aleatoric and epistemic. A widespread consensus is to consider the softmax applied output of a neural network as the confidence of the classifier. However, it is not clear what kind of uncertainty that score captures. Literature in proper scoring rule would suggest that it measures the posterior probability of the input falling into some class, which is inherently an aleatoric notion of uncertainty. However, practitioners and researchers in the community also hope that such a score would be measure of reliability of rare / out-of-distribution / atypical samples, which is more of an epistemic nature of uncertainty. Therefore, there is some sort of conflation is happening on the nature of this score. A sample could be low confidence because it is inherently ambiguous (aleatoric) or is not well represented in the training distribution (epistemic). The paper opens up with this question, and gives arguments to quantify *atypicality* of the sample as well for more reliability. With a simple measure to quantify atypicality, it shows that there is a a correspondence between how atypical a sample is and how mis-calibrated it is. It then studies atypicality aware recalibration. There is a sufficient empirical evaluation.

**Strengths:**

1. One of the major strengths of this paper is that it raises a very relevant question. Interpretation of the confidence outputted by the classifier has a tendency to suffer from serious conflation issues where it is not clear what it captures. The paper argues for a different notion that can settle this issue (or at least start interesting discussions in the community). I believe this is highly valuable.

2. While the operationalisation of the proposed atypicality score is simple, the scoring criteria is already good enough to support the arguments made in the paper in a satisfactory way.

3. Thorough empirical evaluation is done on different settings ranging from balanced supervised classification, imbalanced classification , Language modelling, recalibration, and conformal prediction uncertainty sets on a range of datasets.

**Weaknesses:**

1. There are no major glaring weaknesses of the paper. Except that maybe it underplays itself, as in my mind, it puts interesting and insightful commentary on the nature of softmax output (or sigmoid for Binary classification) of the classifier. There is a great debate on what that is, see Appendix B.2 in [1]. The paper could be even more impactful if these discussions are taken into consideration.

2. Section 3.2: Although some theoretical results are presented that shows correspondence between atypicality and overconfidence in a simple model, it is not clear how these results apply to the other things in the paper. It would be good to draw other insights from it. Do these results provide any insight (or intuition) on why atypical examples would be mis-calibrated? Is this due to estimation errors?

Personally, I thought about this question myself. And to think of it,  it does make sense intuitively why mis-calibration might be an issue for atypical examples. Consider Binary classification, and assume that one is perfectly able to model $\eta(x) = \mathbb{P}(Y=1\vert X=x)$ where the evaluation of $\mathbb{P}$ happens with respect to the training data distribution $P$. Now for a sample $x_{new}$ that is not well-represented in $P$, it is not unsurprising to see that $\mathbb{P}(Y=1 \vert X=x_{new})$ would be low. However, there is no reason to expect that it would be calibrated. Could authors comment on or augment this intuition?




[1] Tim Pearce et al. Understanding Softmax Confidence and Uncertainty. (2021)

**Questions:**

Some questions follow:

1. (Section 4.2): I'm not sure I follow the intuition behind Equation 2. $\psi$ is a function of input atypicality. What is the nature of $\psi$? Is it monotonic, as in it preserves the atypicality of the input. If I assume so, then for a typical input, let's say the atypicality score is 0, in this case $\hat{\mathbb{P}}_{\text{ARR}}(Y\vert X) = \frac{\exp\{S_Y\}}{Z(X)}$, which is not intuitive. I'm sure this is my misunderstanding, so would appreciate the clarification. Specifically, the nature of $\psi$ and $\phi$.

2. (Line 222): The paper shows that atypicality-aware recalibration improves the overall accuracy in imbalanced supervised classification. In my mind, this is problematic, no? Recalibration techniques should preserve the nature of classifier, is called the sharpness / refinement property of calibration maps. Furthermore, ECE roughly translates to difference in average confidence and average accuracy. And the high ECE in atypicality is the result of lower accuracy but higher confidence. One could improve on ECE by compensating for accuracy instead of the confidence, which in my opinion is not the goal of calibration. I would love to hear from the authors on this.

3. (Section 4.3): I understand that atypicality aware recalibration is useful in fairness methods. Although it is an interesting case study and not the major focus of the paper, could authors comment (and compare) on the notion of multicalibration in the literature. I think atypicality aware recalibration has the same spirit as multicalibration in fairness literature.

4. (Section 5, Line 266): For conformal prediction, the authors write "group points according to their confidence and atypicality quantiles...", do the authors use some function to club the confidence score and the atypicality score into a single score? Or some multiple testing procedure is used to accommodate both the score?

---

> ### Author Rebuttal · Authors · 2023-08-09
>
> Dear Reviewer y4KB,
>
> Thank you very much for your detailed review! We really appreciate to hear that you find that the notions proposed in the paper can start interesting discussions in the community and our empirical evaluation to be thorough. We really appreciate your approval and kindness!
>
> Below are our responses to your questions:
>
> ### Response to Questions / Comments
>
> - Q1. When $\psi(a(X))=0$, the output distribution reduces to a fixed distribution over classes that was estimated on the calibration set. This distribution can be seen to induce a prior probability over classes. $\psi$ trades off between this distribution and the model’s confidence. When the point is typical, the output distribution is closer to $\hat{P}(Y|X)$. Empirically, we indeed observe $\phi$ to have a monotonic relationship with atypicality. Note that Theorem 3.1 similarly suggests a monotonic relationship. If a point is atypical (say $\psi \approx 0$), then the linear model is more overconfident, thus we should have a larger correction (e.g. Larger temperature).
>
> - Q2. Thank you for raising this important question. Most of the accuracy improvements arise in the imbalanced classification settings. As we observe in our analyses, the model often puts less probability mass on the rare classes and puts more confidence on the typical(common) classes. If we add class-typicality in the recalibration procedure, we can fix this issue, and this can change(and increase) the accuracy. Indeed, in Appendix Figure 11, we observe that more atypical classes get larger positive corrections, which increases the predicted probability.
>
> - Q3. Indeed there are shared notions between multicalibration and our work. While multicalibration can be considered as a general framework to study group fairness, we are interested in a very specific definition of groups, which are atypical and typical groups. We believe that atypicality would be a useful notion that could be integrated into any multicalibration algorithm, which hopefully future work can study.
>
> - Q4. We split points according to both the atypicality and confidence. Namely, a group $G$ here is defined using 4 thresholds, namely $G = \{x: (l_{a}^{(G)} < q_{a}(x) \leq h_{a}^{(G)}) \land (l_{c}^{(G)} < q_{c}(x) \leq h_{c}^{(G)}) \}$ where $q_{a}(x)$ and $q_{c}(x)$ denote the atypicality and confidence quantiles for the sample $x$, $l_{a}^{(G)}$ and $h_{a}^{(G)}$ denote the atypicality lower and upper bounds for group $G$, and $l_{c}^{(G)}$ and $h_{c}^{(G)}$ denote the confidence lower and upper bounds for group $G$. Using a calibration set, these bounds are simply determined by the quantiles of confidence and atypicality statistics.
>
>
> - On your intuition & Section 3.2: Thank you for providing your intuition! First of all, the insights from the theoretical model indeed transfer empirically to more complex models. As one can note in Figure 1a or Figure 5-6, predictions for more atypical instances are more overconfident and have lower accuracy/coverage(for prediction sets). Further, we mostly agree with your intuition on the estimation errors, but this alone is not sufficient to explain why specifically overconfidence happens (and say, not underconfidence). For this point, the activation function becomes an important factor, and Bai et al. results that we build on provide a good discussion.
>
> - Pearce et al.: This is indeed a very interesting and relevant reference, thank you for bringing this to our attention! We will be sure to include this reference in our revision.
>
> Thank you so much for all your questions and comments, we really appreciate your time! Please let us know if there are any further questions that you may have.

---

> > ### Comment · Reviewer_y4KB · 2023-08-15
> > **Post-rebuttal comment**
> >
> > Thanks to authors for the response, and for the nice Bai et al. reference (I missed it before). The response clarifies my questions. However, I would suggest authors to elaborate on these points (especially Q1 and Q4) in the main text. I do agree with other reviewers that presentation wise paper could be improved. However, I liked the central idea and goal of this paper, and the simplicity with which it goes on achieving it. I'm happy to increase my score as I think this paper raises important points.

---

> > > ### Author Response · Authors · 2023-08-15
> > > **Thank you!**
> > >
> > > Dear Reviewer y4KB,
> > >
> > > We truly appreciate your approval! We will elaborate on the explanations to Q1-Q4 in the main text.
> > >
> > > Thank you again for the time you took for interaction during the discussion phase.

---

### Decision · Program_Chairs · 2023-09-21

**Decision:**

Accept (poster)

**Comment:**

This paper addresses the issue of model confidence in relation to a notion of atypicality, showing that traditional models are often strongly overconfident on examples that are drawn from regions of less support in the training data.  The paper builds on prior work investigating the limitations of model confidence for uncertainty quantification, and proposes that models should incorporate atypicaltiy measures directly when quantifying uncertainty.

After careful reviews and a full set of responses to author rebuttal, there was a clear consensus among the reviewers that the paper is addressing an important topic, making a simple but useful contribution that is well motivated and shown to be strongly effective.  I expect that the authors will incorporate the feedback from the reviewers into the revision for the final paper, especially using the points raised for clarification as an opportunity to strengthen the paper for the general (typical?) NeurIPS reader, but overall the paper is clearly ready for acceptance.

The only thing I would add is that to me, it appears that there is likely a strong connection between the notion of atypicality proposed here and observations around the effectiveness of deep ensembles for uncertainty quantification.  It seems to me that ensemble disagreement is likely to be highest for atypical examples, and that there may be a useful connection there that can help to draw some unified conclusions.